# BACH2 regulates T cell lineage state to enhance CAR T cell function

Tien-Ching Chang [1,2], Amanda Heard [1,2], John Lattin[2,3], John M. Warrington [1,2], Amanda Barrett[1,2], Jack H. Landmann[1,2], Yangdon Tenzin[1,2], Vishaal Ganesh[1,2], Bryant Thompson[1,2], Sadia Afrin[1,2], Deepesh Kumar Gupta[1,2], Ju-Fang Chang[1,2], Julie Ritchey[1,2], Mehmet Emrah Selli [1,2], Yu-Sung Hsu[1,2], Haorui Song [4], A. J. Federico[5], Avery Horn[1,2], Michael P. Meers[5], Evan W. Weber [6], Thomas J. Wandless[7], Jeremy Chase Crawford [8,9], Paul G. Thomas [8,9], John F. DiPersio [1,2], Stephen Gottschalk [10] & Nathan Singh [1,2] ✉

Nearly all chimeric antigen receptors (CARs) signal in the absence of antigen, referred to as 'tonic signaling'. Tonic signaling of CARs containing 41BB domains enhances T cell fitness and function, in contrast to the exhaustion driven by CD28-containing CARs. Here we show that 41BB induces BACH2, a transcriptional regulator that directs stem and memory programs. Overexpression of BACH2 successfully prevented exhaustion but locked CAR T cells in a quiescent state. We linked BACH2 to a degradation domain to tune BACH2, enabling us to prevent exhaustion while enabling potent effector function that broadly enhanced the long-term efficacy of CAR T cells targeting liquid and solid tumors. Through interrogation of clinical CAR products, we further found an association between BACH2 activity and clinical outcomes in patients with leukemia. These data identify a central function for BACH2 in regulating CAR T cell efficacy.

Like T cell receptors, chimeric antigen receptors (CARs) initiate potent intracellular signaling on binding to target antigen. This signaling induces a variety of molecular activities, primary among which are transcriptionally mediated cell-state transitions. T cell lineage states define T cell functionality and thus antigen receptor signaling is central to the generation and maintenance of comprehensive T cell immunity. Several studies have demonstrated that most CARs intrinsically signal in the absence of antigen, referred to as 'tonic' signaling[1,2]. Although the degree of tonic signaling varies by receptor, tonic signaling that occurs during manufacture of T cell products with CARs containing

the CD28 costimulatory domain has been shown to impair T cell function in several solid tumor models[1–4]. This process conceptually and molecularly mirrors the development of T cell exhaustion that results from prolonged T cell receptor stimulation[5].

Previous work showed that shortening of the amino acid linker connecting the single-chain variable fragment (scFv) heavy and light chains induced tonic CAR signaling by forcing CAR clustering[6]. For CARs containing the 41BB costimulatory domain, this tonic signaling enhanced T cell fitness and function in models of blood cancer, suggesting a divergent functional impact of tonic signaling that is dependent on the CAR

[1]Division of Oncology, Section of Cellular Therapies, Washington University School of Medicine, St Louis, MO, USA. [2]Center for Genetic and Cellular Immunotherapy, Washington University School of Medicine, St Louis, MO, USA. [3]Department of Medicine, Washington University School of Medicine, St Louis, MO, USA. [4]Saint Louis University School of Medicine, St Louis, MO, USA. [5]Department of Genetics, Washington University School of Medicine, St Louis, MO, USA. [6]Division of Oncology, Department of Pediatrics, The Children's Hospital of Philadelphia and University of Pennsylvania School of Medicine, Philadelphia, PA, USA. [7]Department of Chemical and Systems Biology, Stanford University, Stanford, CA, USA. [8]Department of Host-Microbe Interactions, St Jude Children's Research Hospital, Memphis, TN, USA. [9]Center for Infectious Diseases Research, St Jude Children's Research Hospital, Memphis, TN, USA. [10]Department of Bone Marrow Transplantation and Cellular Therapy, St Jude Children's Research Hospital, Memphis, TN, USA. ✉e-mail: nathan.singh@wustl.edu

costimulatory domain. These findings are consistent with two clinical trials of 41BB-containing CD22 CAR T cell products, in which tonically signaling CARs induced remission in >75% of patients[7], whereas CARs that did not signal tonically only mediated responses in ~20% (ref. [6]). Tonic 41BB signaling likely does not occur in natural T cell responses and, based on its apparent beneficial impact on engineered T cell function, we were motivated to understand the responsible molecular circuits.

Using CARs targeting CD22, we confirmed the contrasting effect of tonic CD28 and 41BB signaling on CAR T cell function. We identified that tonic 41BB-based CAR signaling activates BACH2, recently shown to promote quiescent stem and memory-like T cell lineages[8,9]. Through a series of in vitro studies, we demonstrated that the quantity of BACH2 present during CAR T cell manufacturing has a critical effect on CAR T cell lineage identity and function. We further defined an approach to dynamically tune BACH2 to enhance the function of CAR T cells targeting liquid and solid tumors in vivo. Finally, we found that clinical CAR T cell products with higher BACH2 activity are associated with enhanced responses in patients with acute lymphoblastic leukemia (ALL). Collectively, these data demonstrate that BACH2 is a core regulator of CAR T cell lineage and function and that synthetic control of BACH2 quantity can balance the rheostat controlling exhausted, memory and effector states to enhance long-term efficacy.

## Results

### CAR costimulatory domains control the effect of tonic signaling

To first determine whether the observed divergent impact of tonic signaling from CD28 (refs. [1–4]) and 41BB-based[6] CARs was an artifact of distinct CAR/target models, we established a controlled system to test the impact of tonic costimulatory domain signaling on T cell fitness. We used scFv linker length to control tonic signaling[6], synthesizing four CARs containing αCD22-binding domains from the m971 antibody[10] with either long (20 residues, no tonic signaling) or short (5 residues, tonic signaling) linkers and containing either CD28 or 41BB costimulatory domains (Fig. 1a; receptors will be referred to as 22/BB-L (41BB domain, long linker), 22/BB-S, 22/28-L or 22/28-S). We found that expression of 22/BB-S induced the best T cell expansion during product manufacturing, that long-linker CARs were equivalent to untransduced controls (NTDs) and that 22/28-S impaired expansion (Fig. 1b). We further noted that 22/28-S drove higher expression of the exhaustion marker PD1 (Fig. 1c). 22/BB-S cells were more effective at killing CD22+ Nalm6 leukemia cells in vitro than their nontonic counterpart 22/BB-L cells (as we previously showed[6]) and 22/28-S induced only modest Nalm6 killing compared to controls (Fig. 1d). Measurement of core effector cytokines interferon γ, interleukin-2 and tumor necrosis factor revealed consistent trends (Fig. 1e), as did CAR T cell expansion in response to tumor engagement (Fig. 1f). We observed robust and maintained expression of PD1, TIM3 and LAG3 by 22/28-S (Fig. 1g) which, in the context of impaired function, we conclude to reflect T cell exhaustion. These data confirm that tonic CD28-based CAR signaling is detrimental, whereas tonic 41BB-based CAR signaling is beneficial to CAR T cell function.

### Signaling of 22/BB-S induces BACH2

To determine the molecular etiology of this contrasting impact of tonic CAR signaling, we interrogated differences in gene expression using RNA sequencing (RNA-seq). To amplify the tonic signal and reduce the 'noise' of αCD3/CD28 bead stimulation, we collected CAR T cells at the conclusion of manufacture. Analysis of differentially expressed genes (DEGs) demonstrated broadly increased gene expression in short linker compared to long linker CAR T cells, consistent with antigen-independent activation (Fig. 2a). Genes with higher expression in 22/BB-S compared to 22/28-S largely consisted of memory lineage-associated genes (such as *CCR7*, *TCF7* and *KLF2*), consistent with 41BB's known role in directing memory differentiation[11]. We also saw

that 22/BB-S drove expression of class II major histocompatibility complex (MHC) genes, an observation that we and others have previously made in other models of 41BB-based CAR activation[12,13]. Gene set enrichment analysis (GSEA) of genes upregulated in 22/BB-S revealed robust enrichment of genes regulated by BACH2 (Fig. 2b), a transcriptional repressor critical in dictating fate decisions in B cells[14] and recently shown to promote naive, stem and memory-like lineages in T cells[8,9]. Similar analysis for 22/28-S demonstrated high activity of NFAT and AP-1, T cell activating factors that, when dysregulated or active for prolonged periods, can drive T cell exhaustion[3,15] (Extended Data Fig. 1a). We confirmed that high NFAT and AP-1 activity was driven by resting 22/28-S in Jurkat reporters (Extended Data Fig. 1b). We then assayed differences in accessibility of BACH2-binding sites using the assay for transposase-accessible chromatin with sequencing (ATAC−seq) and conducted transcription factor motif analysis (using chromVAR). These data confirmed a significant enrichment of open BACH2-binding motifs in 22/BB-S which was lost with *BACH2* disruption (Fig. 2c and Extended Data Fig. 1c).

To further confirm higher BACH2 activity in 22/BB-S, we interrogated DEGs specifically between 22/BB-S and 22/28-S for expression of known BACH2 targets[16], which demonstrated enrichment in 22/BB-S (Fig. 2d). Consistent with this, we found that 22/BB-S cells had higher quantities of both *BACH2* transcripts and BACH2 protein at the end of manufacture (Fig. 2e and Extended Data Fig. 1d). To determine when during manufacture BACH2 expression was induced, we quantified the relative change in BACH2 protein from the day after bead removal (day 7) to the conclusion of manufacture (day 14). We found almost no change in 22/28-S or control T cells but an ~4× increase in BACH2 in 22/BB-S, suggesting that 22/BB-S activity was itself responsible for inducing BACH2 expression (Fig. 2f). BACH2 is regulated by both gene expression and post-translational modification; active BACH2 is intranuclear but its phosphorylation results in segregation to the cytoplasm[9]. Confocal microscopy quantifying BACH2's cellular compartmentalization demonstrated that 22/BB-S had the highest nuclear to total BACH2 ratio (Fig. 2g,h). Collectively, these data comprehensively identify induction of BACH2 by 22/BB-S.

### BACH2 redirects lineage states to enhance CAR T cell function

Based on BACH2's known function, we hypothesized that BACH2 was responsible for the enhanced function of 22/BB-S and could enhance the function of 22/28-S. BACH2 is large (92 kDa) and thus we elected to engineer expression of CAR and BACH2 on two independent lentiviral vectors (each with independent transduction markers, truncated CD34 for CAR and truncated EGFR for BACH2; Fig. 3a). We observed robust expression of BACH2 in tEGFR+ human T cells (Fig. 3b) which was equivalent in CD4 and CD8 T cells (Extended Data Fig. 2a). At the conclusion of manufacturing, 22/28-S+BACH2+ cells had significantly lower expression of PD1 (Fig. 3c), LAG3 and TIM3 (Extended Data Fig. 2b) than 22/28-S cells.

We next evaluated the impact of BACH2 on memory lineage. Central memory T cells (T_CM cells) are an 'early' lineage state responsible for long-term memory with high functional potential (capability of self-renewing and differentiating into cells with high function) but limited direct functionality[17]. Effector memory T cells (T_EM cells), in contrast, are shorter lived with lower long-term potential but potent immediate function. We found that overexpression of BACH2 (BACH2^OE) resulted in a significant increase in $CD62L^+CD45RO^+$ T_CM cells with a concomitant reduction in $CD62L^-CD45RO^+$ T_EM cells in both CD4 and CD8 lineages (Fig. 3d and Extended Data Fig. 2c). We observed the same trend when we overexpressed BACH2 in T cells expressing a CD19 CAR that does not signal tonically and a CD33 CAR that does signal tonically[6] (Extended Data Fig. 2d). Broader phenotypic evaluation using mass cytometry (Supplementary Table 1) revealed that BACH2^OE induced a generalized transition in 22/28-S to a 22/BB-S-like phenotype (Fig. 3e). RNA-seq at the conclusion of T cell manufacture confirmed that BACH2^OE significantly altered gene

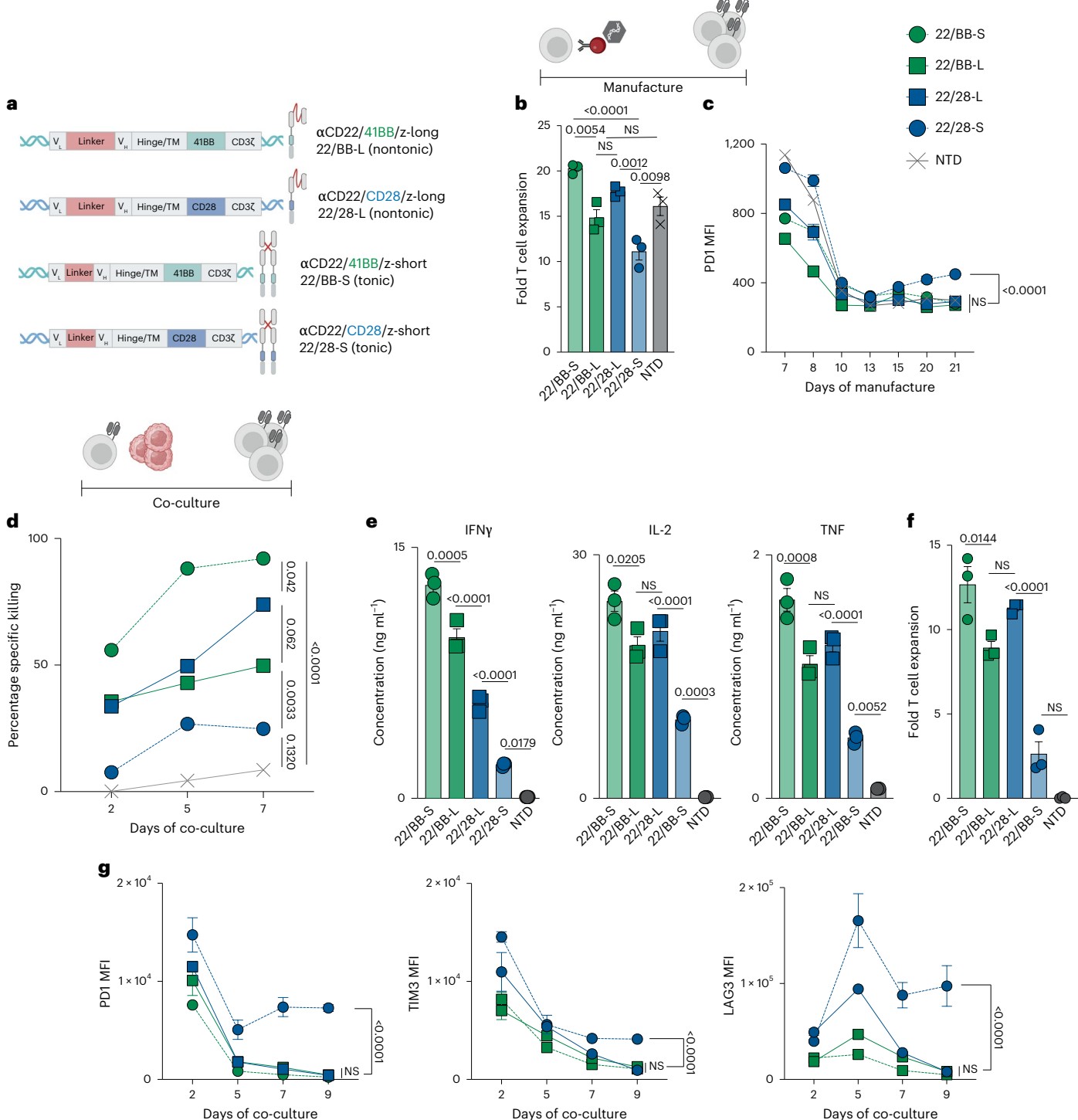

**Fig. 1 | CAR costimulatory domains control functional effect of tonic signaling.**
**a**, Schematic of the four CAR constructs generated to test the impact of tonic signaling. **b**, CAR T cell expansion during manufacture (NTDs). **c**, PD1 expression during CAR T cell manufacture. **d**, Nalm6 cell survival over time in co-culture with CAR T cells. **e**, Quantification of cytokines in culture supernatants 48 h after combination of Nalm6 and CAR T cells. **f**, CAR T cell expansion at day 6 of co-culture with Nalm6. **g**, Expression of PD1, TIM3 and LAG3 by CAR T cells over the course of co-culture with Nalm6. For **b**–**g**, representative data are from *n* = 4 donors. All data are presented as either individual values or mean values ± s.e.m. *P* < 0.05, *P* < 0.01, *P* < 0.001, *P* < 0.0001 by two-way analysis of variance (ANOVA) with Bonferroni's adjustment for multiple comparisons. IFNγ, interferon γ; IL-2, interleukin-2; MFI, mean fluorescence intensity; NS, not significant; TNF, tumor necrosis factor. Schematic in **a** created with BioRender.com.

expression in 22/28-S (Fig. 3f). Identification of DEGs across groups revealed that 22/BB-S maintained high expression of class II MHC genes (regulated by the factor CIITA, the second-most active network in these cells (Fig. 2b)), whereas 22/28-S⁺BACH2ᴼᴱ cells were now the cell type most enriched for expression of memory genes (Fig. 3g).

GSEA using the ImmuneSig database[18] demonstrated that, compared to 22/28-S, 22/28-S⁺BACH2ᴼᴱ cells were highly enriched for naive and memory lineages (Fig. 3h). Orthogonal validation using gene set variation analysis[19] confirmed enrichment of naive signatures and suppressed effector and exhausted signatures (Fig. 3i).

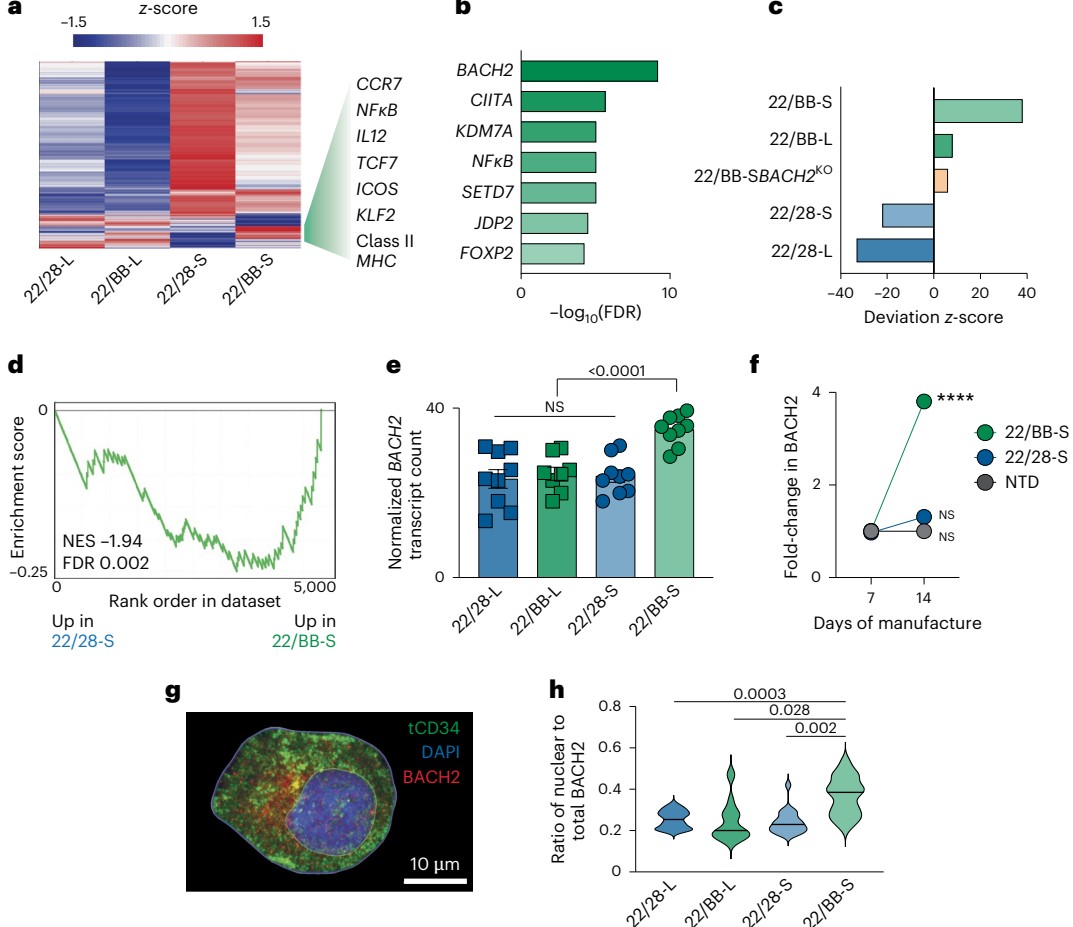

**Fig. 2 | Tonic 41BB signaling induces BACH2. a**, Heatmap of DEGs (log$_2$(fold-change) >1.5, false discovery rate (FDR) <0.05) across 22/BB-S, 22/28-S, 22/BB-L and 22/28-L at the conclusion of CAR T cell manufacture. Pooled data from $n = 3$ donors were performed in technical triplicates. **b**, Transcriptional regulators of genes significantly upregulated by 22/BB-S, identified by GSEA. **c**, ChromVAR analysis of ATAC–seq data evaluating BACH2 motif accessibility. **d**, GSEA of BACH2 target gene expression in genes differentially expressed by 22/BB-S and 22/28-S. **e**, Normalized *BACH2* transcript counts in CAR T cells at the end of manufacture ($n = 3$ donors). **f**, Fold-change in BACH2 protein expression over the course of CAR T cell manufacture (representing $n = 2$ donors). **g**, Representative image of CAR T cell stained for tCD34 (transduction marker), BACH2 and DAPI with boundaries drawn around the whole cell and nucleus. **h**, Ratio of nuclear to total BACH2 quantified by confocal microscopy ($n = 20$ cells per group). All data are presented as either individual values or mean values ± s.e.m. $P < 0.05$, $P < 0.01$, $P < 0.001$, ****$P < 0.0001$ by two-way ANOVA. NES, normalized enrichment score.

---

We next conducted a series of functional analyses, which revealed that BACH2$^{OE}$ mediated complete rescue of 22/28-S cytotoxicity, enabling similar tumor control to 22/BB-S (Fig. 3j). This was accompanied by an increase in T cell expansion and a significant reduction in PD1 expression (Fig. 3k,l). We observed that BACH2$^{OE}$ promoted expression of TCF1, necessary for directing T$_{CM}$ cell formation[14,18], and restrained expression of Eomes, critical for transition and maintenance of T$_{EM}$ cells[20,21] (Fig. 3m). In light of the profound similarity in phenotype and function between 22/BB-S and 22/28-S$^+$BACH2$^{OE}$, we hypothesized that BACH2 activity was central to the identity and superior function of 22/BB-S cells. Disruption of *BACH2* in 22/BB-S (Fig. 3n) was associated with a significant decline in T$_{CM}$ cells and increase in T$_{EM}$ cell compartments at the end of CAR T cell manufacture (Fig. 3o), as well as a reduction in T cell function after short-term (Fig. 3p,q) and long-term antigen stimulation (Extended Data Fig. 2e). Collectively, these studies identify a central role for BACH2 in regulating CAR T cell lineage and function and confirm a direct link of tonic 41BB signaling, BACH2 induction and enhanced T cell fitness.

## Overexpression of BACH2 prevents transit to effector states

We next evaluated whether BACH2$^{OE}$ extended long-term CAR T cell function, specifically T cell expansion and persistence, in response to a chronic antigen load. We employed a serial re-stimulation assay in which manufactured CAR T cell products are co-cultured with targets at higher effector-to-target (E-to-T) ratios (1:4) and continually re-fed with fresh Nalm6 to maintain a consistently high antigen burden[22]. We again observed that BACH2$^{OE}$ initially enhanced T cell function (measured here by T cell expansion) but caused an abrupt T cell contraction (Fig. 4a). Lineage profiling revealed that, in contrast to 22/BB-S and 22/28-S, BACH2$^{OE}$ maintained T$_{CM}$ cell phenotypes over time and impaired transit into T$_{EM}$ cell states in response to prolonged antigen exposure for both CD4s and CD8s (Fig. 4b,c and Extended Data Fig. 3a). Notably, we observed that CD4s were enriched for T$_{CM}$ cell states at the expense of T$_{EM}$ cells in 22/BB-S and 22/28-S$^+$BACH2$^{OE}$ cells, although this enrichment was modest (maximum difference of 8% between CD4 and CD8 T$_{CM}$ cells; Extended Data Fig. 3b). The BACH2 quantity in CD4 and CD8s cells remained equivalent (Extended Data Fig. 3c). BACH2$^{OE}$ cells failed to substantively upregulate Eomes or downregulate TCF1 (Fig. 4d,e). The 22/28-S$^+$BACH2$^{OE}$ cells had reduced co-expression of KLRG1 and granzyme B (GZMB), an indicator of activated cells transiting from T$_{CM}$ cells to T$_{EM}$ cells[23,24] (Fig. 4f). Directed examination of memory or cytotoxicity markers CD127 or KLRG1 and TCF1 or GZMB confirmed these lineage patterns and we observed that, although memory markers were similar for CD4 and CD8 cells, CD8 cells expressed higher levels

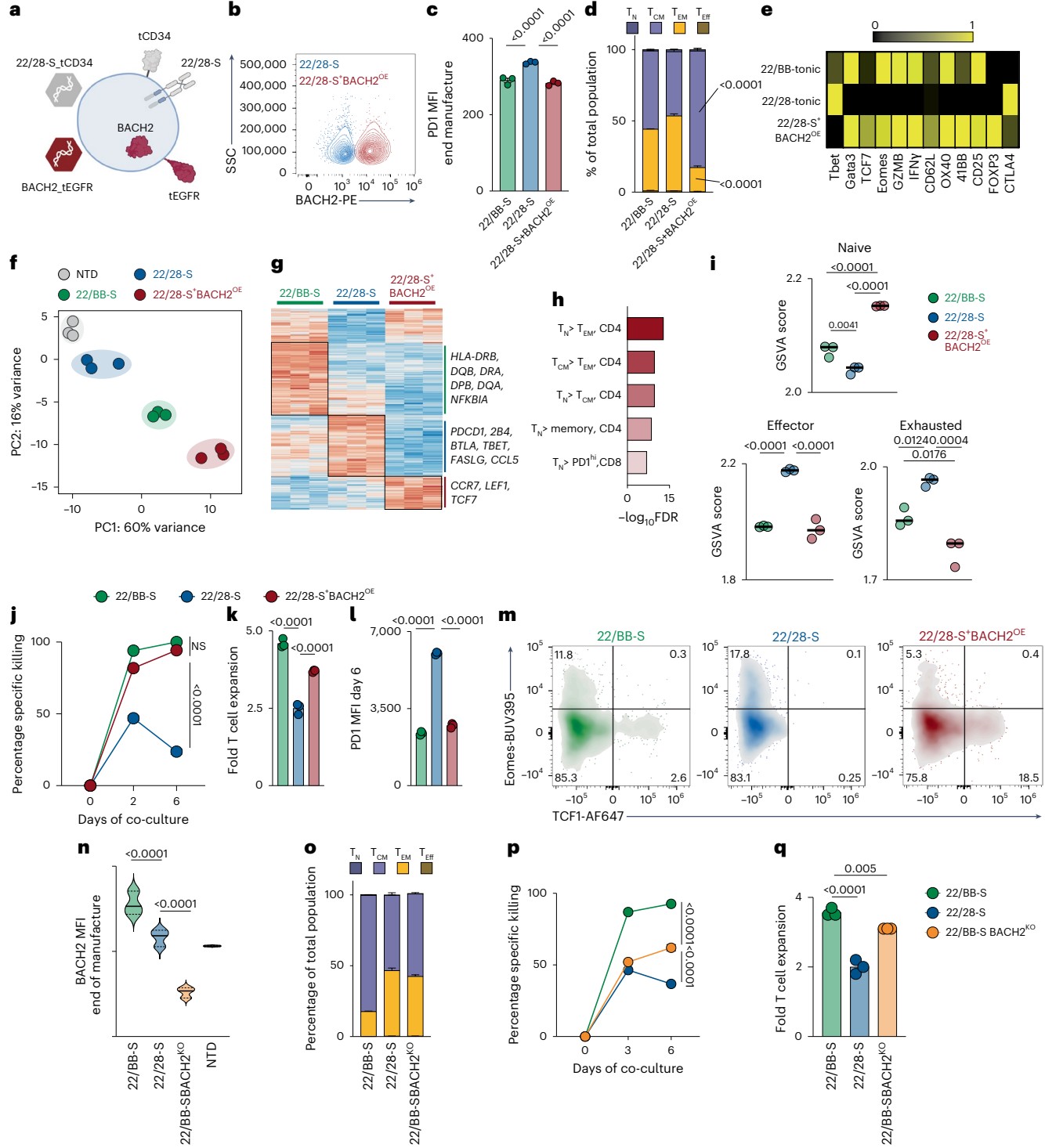

**Fig. 3 | BACH2^OE enhances function of 22/28-S. a**, Schematic of engineering approach using two lentiviral vectors with distinct transduction markers. **b**, Expression of transgenic BACH2. **c**, PD1 expression at the end of CAR T cell manufacture. **d**, Memory subset composition of CAR products at the end of manufacture. Subsets defined as naive cells (T_N cells, CD62L^+CD45RO^−), T_CM cells (CD62L^+CD45RO^+), T_EM cells (CD62L^−CD45RO^+) or terminal effector (T_Eff, CD62L^−CD45RO^−) cells. **c,d**, Representing $n = 4$ donors. **e**, Normalized expression of lineage-defining proteins quantified by mass cytometry. **f**, Principal component analysis representing variance in gene expression of CAR T cells at the end of manufacturing ($n = 1$ donor performed in technical triplicate). **g**, Heatmap of DEGs across 22/BB-S, 22/28-S and 22/28-S^+BACH2^OE. **h**, Lineage signatures of genes upregulated in 22/28-S^+BACH2^OE compared to 22/28-S as defined in the ImmuneSig database. **i**, Gene set variation analysis (GSVA) of 22/BB-S, 22/28-S and 22/28-S^+BACH2^OE using naive, effector and

exhausted signatures. **j**, Survival of Nalm6 over time in co-cultures with CAR T cells. **k**, Fold CAR T cell expansion on day 6 of co-culture with Nalm6. **l**, CAR T cell PD1 expression on day 6 of co-culture with Nalm6. **m**, Representative flow cytometry plots of co-expression of Eomes and TCF1 on day 3 of co-culture. For **j–m**, data represent $n = 4$ donors. **n**, BACH2 protein expression after CRISPR-mediated *BACH2* disruption in CAR T cells. **o**, Memory subset composition of CAR T cell products at the end of manufacture. **p**, Nalm6 survival over time in co-culture with CAR T cells. **q**, Fold CAR T cell expansion on day 6 of co-culture with Nalm6. For **n–q**, representative data are from $n = 2$ donors. All data are presented as either individual values or mean values ± s.e.m. $P < 0.05$, $P < 0.01$, $P < 0.001$, $P < 0.0001$ by two-way ANOVA with Bonferroni's adjustment for multiple comparisons. PC, principal component. Schematic in **a** created with BioRender.com.

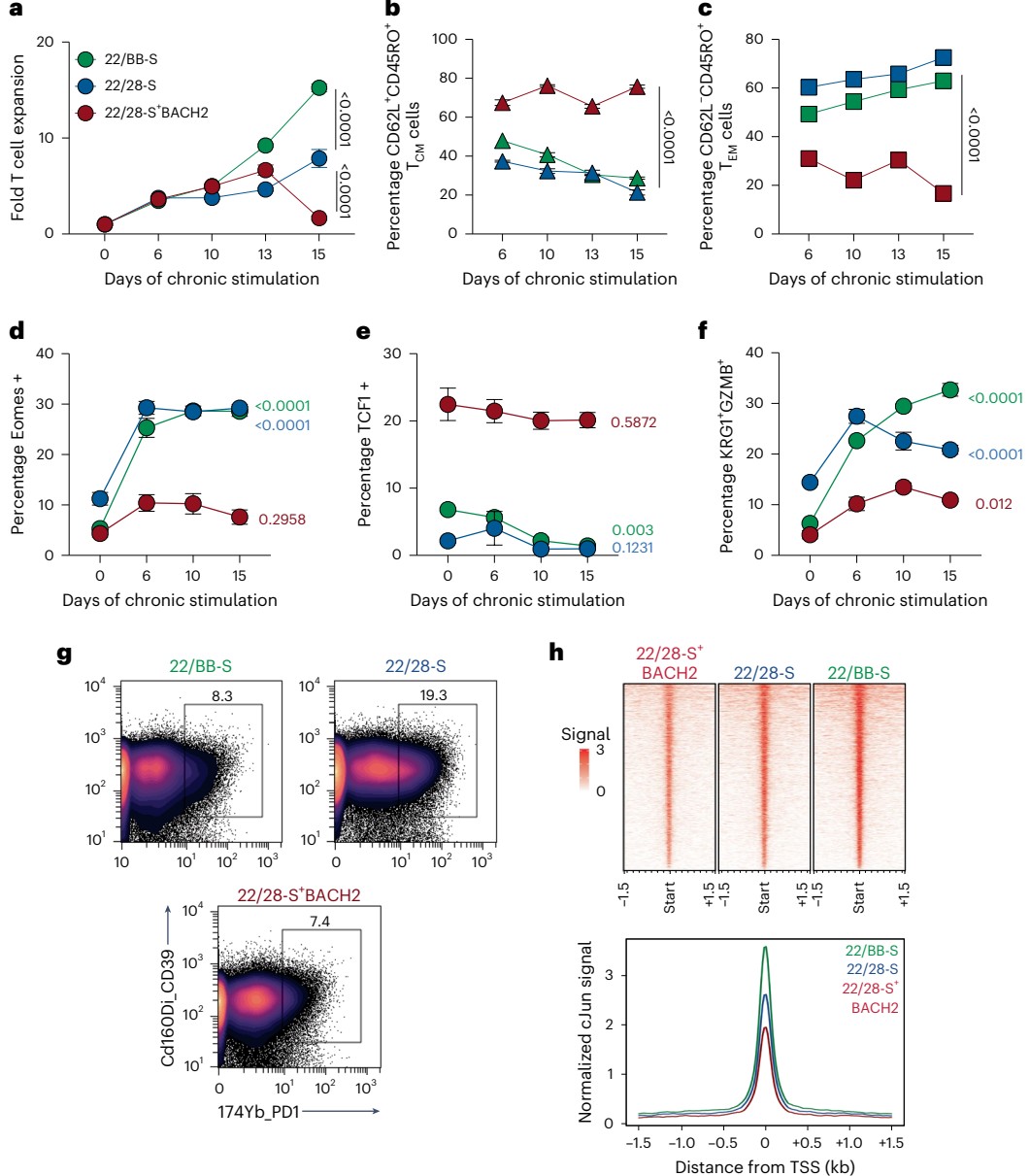

**Fig. 4 | BACH2^OE restrains long-term function of 22/28-S by preventing differentiation. a**, Fold expansion of CAR T cells in chronic stimulation co-cultures. **b,c**, Central (**b**) and effector (**c**) memory composition of CAR T cells over time. **d**, Expression of Eomes. **e,f**, TCF1 (**e**) and co-expression of KLRG1 and GZMB (**f**) over time. For **a**–**f**, representative data are from $n$ = 5 independent donors. **g**, Representative flow cytometry plots of CD39 and PD1 co-expression at day 13 of co-culture. They represent $n$ = 3 donors. **h**, CUT&RUN analysis of cJun binding by heatmap (top) and average binding histogram (bottom). All data are presented as either individual values or mean values ± s.e.m. $P < 0.05, P < 0.01$, $P < 0.001, P < 0.0001$ by two-way ANOVA with Bonferroni's adjustment for multiple comparisons. TSS, transcription start site.

of cytotoxicity markers (Extended Data Fig. 4). Notably, these cells did not express high levels of CD39 and PD1 (Fig. 4g), indicating that the observed failure was not a result of exhaustion but that constitutive BACH2^OE decreases T cell transit into the effector states necessary for long-term activity.

In murine models of viral infection, BACH2 has been shown to maintain T cell quiescence by occupying genomic targets of activating AP-1 transcription factors and preventing their activity[9]. Using CUT&RUN analysis, we found that BACH2^OE reduced binding of cJun, part of the AP-1 heterodimer cJun–cFos that is critical for T cell effector function (Fig. 4h), implicating a mechanism for how BACH2^OE restrains long-term CAR T cell efficacy. Intriguingly, we also observed that 22/BB-S had higher binding of cJun–cFos, consistent with its enhanced function.

## Tuning BACH2 enhances long-term CAR T cell function

We hypothesized that this lineage restriction was a result of maintained BACH2 expression throughout long-term engagement with tumor. To test this, we linked transgenic BACH2 to a degradation domain (DD) derived from *Escherichia coli* dihydrofolate reductase (ecDHFR)[25]. This domain compels protein degradation in the absence of a stabilizing ligand, thus enabling us to titrate BACH2 expression using the DHFR antagonist trimethoprim (TMP), a common antibiotic used in humans (Fig. 5a). We engineered co-expression of 22/28-S cells and BACH2^DD in either the presence ('ON') or the absence ('OFF') of TMP during manufacture. We found that BACH2^DD ON cells had similar BACH2 quantity to BACH2^OE, confirming that our DD could successfully mimic transgenic BACH2 overexpression (Fig. 5b). We also found that, although BACH2^DD OFF had significantly reduced BACH2 compared to BACH2^DD ON and

BACH2$^{OE}$, the BACH2 quantity was higher than controls, indicating leakiness of the DD system. Thus, our OFF condition was not truly depleted of transgenic BACH2 but rather 'BACH2$^{low}$', allowing us to simultaneously interrogate how BACH2 quantity (low and high) and kinetics (high during manufacture or high always) impacted CAR T cell function.

We confirmed that the BACH2 quantity was sensitive to the TMP dose, that BACH2 was rapidly degraded on TMP withdrawal and that BACH2$^{DD}$ ON and OFF cells had equivalent BACH2 after withdrawal (Extended Data Fig. 5a,b). All three conditions (ON, OFF and OE) led to robust T$_{CM}$ cell differentiation at the end of manufacture (Fig. 5c) which did not differ for CD4 or CD8 cells (Extended Data Fig. 5c), indicating that even low levels of BACH2 were sufficient to drive this lineage. Similarly, all three conditions reduced PD1 expression (Extended Data Fig. 5d), indicating an ability to prevent exhaustion during manufacture. We found that BACH2$^{DD}$ ON cells had similar long-term functionality to BACH2$^{OE}$ cells, but BACH2$^{DD}$ OFF cells demonstrated significantly enhanced and sustained T cell expansion (Fig. 5d). Tracing of CD4 to CD8 lineages over time revealed that all three BACH2 overexpression conditions enriched CD4s but this ratio slowly normalized over time (Extended Data Fig. 5e). Although all three conditions prevented the development of exhaustion during chronic stimulation (Fig. 5e), BACH2$^{DD}$ OFF cells were able to transit from T$_{CM}$ cells to T$_{EM}$ cells much more effectively than BACH2$^{DD}$ ON or BACH2$^{OE}$ cells (Fig. 5f). Importantly, all three conditions demonstrated minimal terminal effector differentiation (CD62L$^-$CD45RO$^-$; Extended Data Fig. 5f). Consistent with these lineage transitions, BACH2$^{DD}$ OFF cells more efficiently lost TCF1 and gained Eomes expression (Fig. 5g,h). Intriguingly, we found strong direct and indirect correlations between BACH2 quantity and expression of TCF1 and Eomes, respectively, suggesting that BACH2 may serve a higher-order regulatory function in orchestrating lineage-directing transcription factors (Extended Data Fig. 5g,h). BACH2$^{DD}$ OFF cells were also more activated, as measured by co-expression of KLRG1$^+$GZMB$^+$ (Fig. 5i), GZMB alone (Extended Data Fig. 5i) and tracing of CD127 or KLRG1 and TCF1 or GZMB co-expression (Extended Data Fig. 6), further evidence of transit to T$_{EM}$ cell lineage. Using titrated amounts of TMP, we found that the regulatory activity of BACH2 was dependent on BACH2 dose, with higher quantities of BACH2 progressively impairing T cell expansion and locking the central memory state (Extended Data Fig. 7a–d). Collectively, these data reveal that both low and high quantities of BACH2 prevent exhaustion during product manufacture, but that higher expression of BACH2 during manufacture restrains long-term CAR T cell function by impairing cell-state transitions.

We next evaluated how manipulation of BACH2 influenced anti-tumor efficacy compared to 22/BB-S and 22/28-S cells in a xenograft model of human ALL, in which we established systemic disease by intravenous injection of Nalm6 (Fig. 5j). To mimic a high and prolonged antigen burden we conducted a 'stress' test[26], allowing Nalm6 to progress longer than usual (10 d) before infusing a subtherapeutic dose of CAR T cells ($2 \times 10^6$). As for in vitro studies, TMP was given only to CAR T cells during manufacture and not after infusion in the mice. Although expectedly no animals were cured, we observed 22/28-S$^+$BACH2$^{DD}$ OFF and 22/BB-S cells to most effectively delay disease progression (Fig. 5k and Extended Data Fig. 7e). This was associated with a significant improvement in animal survival, with median survival of 26 d for mice receiving 22/28-S cells and 40 d for mice receiving either 22/BB-S or 22/28-S$^+$BACH2$^{DD}$ OFF cells (Fig. 5l; $P < 0.0001$). We profiled splenic CAR T cells at days 7 and 14 after transfer and observed the predicted decline in BACH2$^{DD}$ ON cells after transfer (Fig. 5m). The 22/BB-S cells demonstrated significantly greater expansion and persistence than all other groups and we saw a trend toward increased persistence of BACH2$^{DD}$ OFF cells (Fig. 5n). Memory phenotyping revealed the same trends that we had seen in vitro: a stepwise maintenance of T$_{CM}$ cells and restrained T$_{EM}$ cell transit for BACH2$^{OE}$, followed by BACH2$^{DD}$ ON and then OFF (Fig. 5o,p). We again saw a skewing toward CD4 cells by transgenic BACH2 which was lost over time in vivo (Extended Data Fig. 7f). Collectively, these data suggest that high expression of BACH2 during manufacture alone is sufficient to limit long-term T cell function and that low-level expression of BACH2 compels similar efficacy driven by 22/BB-S.

## BACH2 can overcome highly toxic tonic signaling

The paradigm of tonic signaling is a high-affinity CAR targeting the solid tumor antigen GD2 (refs. 3,4) (HA-GD2), extensively shown to drive the rapid onset of T cell exhaustion[2–4] (Fig. 6a). We sought to determine whether BACH2 could alter the developmental trajectory of T cells expressing this highly toxic CAR. Expression of all BACH2 vectors (BACH2$^{DD}$ ON, OFF or BACH2$^{OE}$) significantly decreased expression of exhaustion markers TIM3 and LAG3 during product manufacture (Fig. 6b) and significantly enhanced T cell function compared to HA-GD2 when chronically exposed to the GD2$^+$ human neuroblastoma cell line SY5Y (Fig. 6c). In contrast to the CD22 model, BACH2$^{DD}$ ON and BACH2$^{OE}$ boosted T cell expansion and prolonged persistence much more effectively in this model. Like 22/28-S$^+$BACH2$^{DD}$ OFF cells, HA-GD2$^+$BACH2$^{DD}$ OFF cells transitioned from T$_{CM}$ cells to T$_{EM}$ cells (Fig. 6d,e); however, low-level BACH2 from BACH2$^{DD}$ OFF did not prevent terminal differentiation or acquisition of CD39 and PD1 (Fig. 6f,g).

Evaluation of these products in vivo confirmed these findings, demonstrating significantly improved tumor control by BACH2-enhanced products with slightly superior function of cells exposed to higher BACH2 (Fig. 6h and Extended Data Fig. 8a). To our surprise, we observed that mice receiving products with transgenic BACH2 began losing weight approximately 2 weeks after T cell delivery, requiring sacrifice beginning on day 26. Changes in weight were quite stark by day 30 (Fig. 6i) and at the day of animal sacrifice (Extended Data Fig. 8b). All mice in control and HA-GD2 CAR groups were sacrificed for disease burden and all mice in HA-GD2$^+$BACH2 groups were sacrificed for weight loss (>15%). Notably these mice did not manifest any other signs of xenogeneic graft-versus-host disease (GVHD) and, at the time of necropsy, had no visible evidence of organ damage. The only abnormal findings were significantly smaller spleen size (Extended Data Fig. 8c) and almost complete loss of intestinal contents, suggestive of anorexia as a cause of weight loss. We isolated spleens from all mice and found fewer total cells in BACH2 groups (Fig. 6j), but that a greater fraction of these cells were CAR$^+$ (Fig. 6k). We found a modest correlation between the fraction of splenic CAR$^+$ T cells and the percentage weight loss (Extended Data Fig. 8d). Phenotypic examination of these CAR$^+$ T cells demonstrated similar trends to our in vitro studies for both GD2 and CD22 models: a progressively increased quantity of T$_{CM}$ cells and a reduced quantity of T$_{EM}$ cells as the BACH2 dose increased (Fig. 6l). These data demonstrate that BACH2 can overcome high levels of detrimental tonic signaling to enhance T cell function and that the quantity of BACH2 needed to overcome tonic signaling-driven exhaustion depends on the strength of the exhaustion signal.

## BACH2 activity is associated with risk of leukemia relapse

Enrichment of early lineage cells in CAR T cell products is associated with enhanced clinical efficacy[27–30], yet the regulatory mechanisms that drive this association are unclear. Given that clinical experience to date is predominantly with CD19 and B cell maturation antigen CARs, we interrogated single-cell RNA-seq (scRNA-seq) data of 12 pre-infusion, CD19-directed, 41BB-based, CAR T cell products[31] manufactured for children with relapsed or refractory ALL. Cellular annotation[32] indicated that the vast majority (>90%) of CAR$^+$ T cells in these products were either or T$_{CM}$ or T$_{EM}$ cells that did not cluster by response status (Fig. 7a) or product identity (Extended Data Fig. 9a). Measurement of BACH2 target gene expression (BACH2 regulon[33]) in these 114,789 cells demonstrated significantly greater BACH2 activity in T$_{CM}$ cells compared to T$_{EM}$ cells (Fig. 7b), consistent with our in vitro

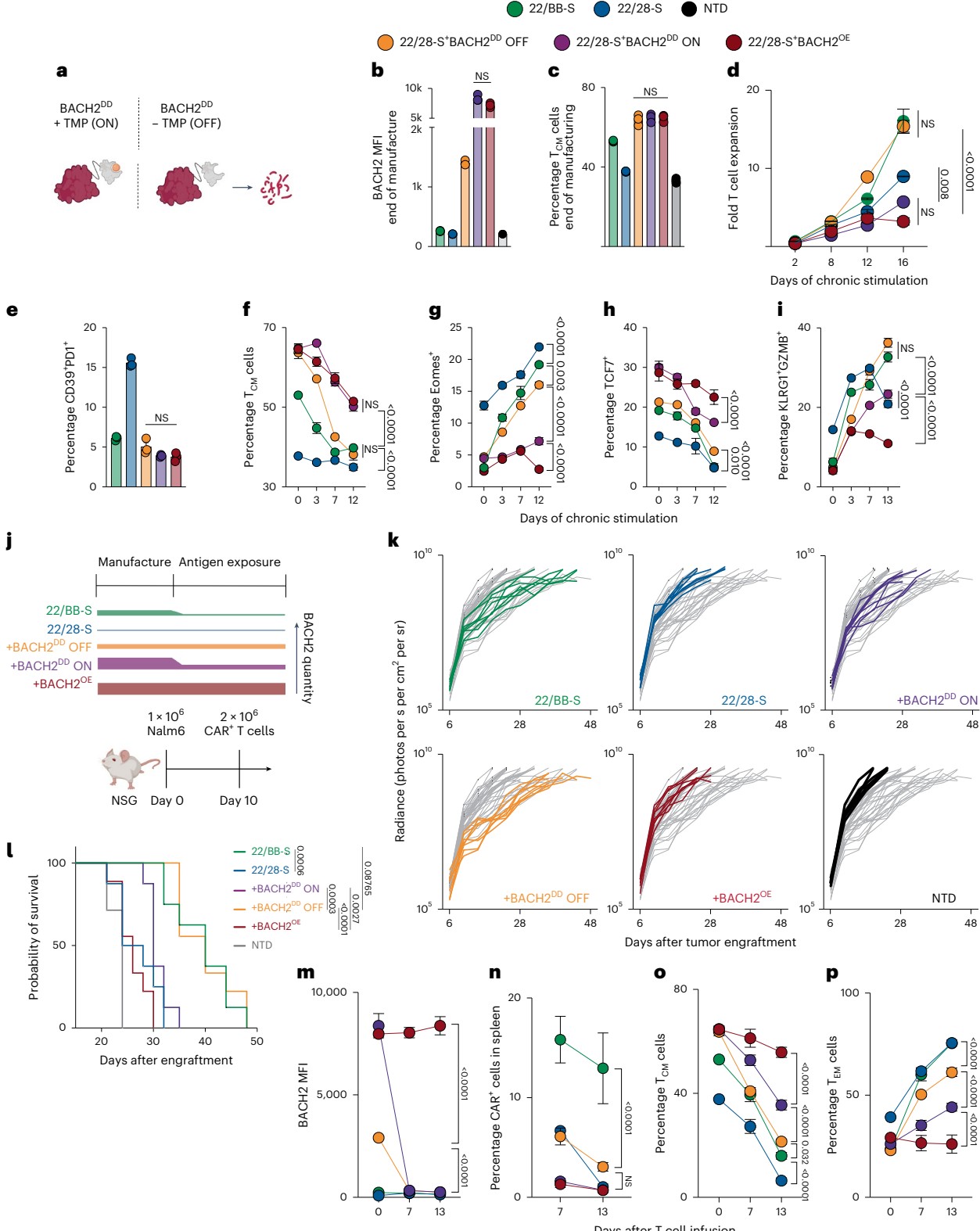

**Fig. 5 | Tuned expression of BACH2 enhances long-term function. a**, Schematic of ecDHFR degron functionality. **b**, BACH2 expression in CAR T cells at the end of manufacture. **c**, Central memory composition of CAR products at the end of manufacture. **d**, CAR T cell expansion over time during chronic stimulation. **e**, Co-expression of CD39 and PD1 over time. **f**, Central memory composition of CAR T cells over time during chronic stimulation. **g**–**i**, Expression of Eomes (**g**), TCF7 (**h**) and co-expression of KLRG1 and GZMB (**i**) over time. Representative data from $n = 3$ donors. **j**, Schematic of BACH2 quantities in various CAR T cell products over the course of manufacture and chronic antigen exposure. **k**, Nalm6

burden over time in mice receiving CAR T cells. **l**, Survival of mice receiving CAR T cells. **k**,**l**, There were $n = 8$ mice per group except for NTD control ($n = 6$). **m**, BACH2 expression in splenic CAR T cells over time. **n**, Percentage of total splenocytes that were CAR+. Percentage of splenic CAR+ T cells that were $T_{CM}$ cells (**o**) or $T_{EM}$ cells (**p**). In **m**–**p**, there were $n = 10$ mice per group, 5 mice evaluated at each time point after infusion. All data are represented as either individual values or mean values ± s.e.m. $P < 0.05$, $P < 0.01$, $P < 0.001$, $P < 0.0001$ by two-way ANOVA with Bonferroni's adjustment for multiple comparisons (**b**–**l**, **m**–**p**) or log(rank test) (**l**). Schematic in **a** created with BioRender.com.

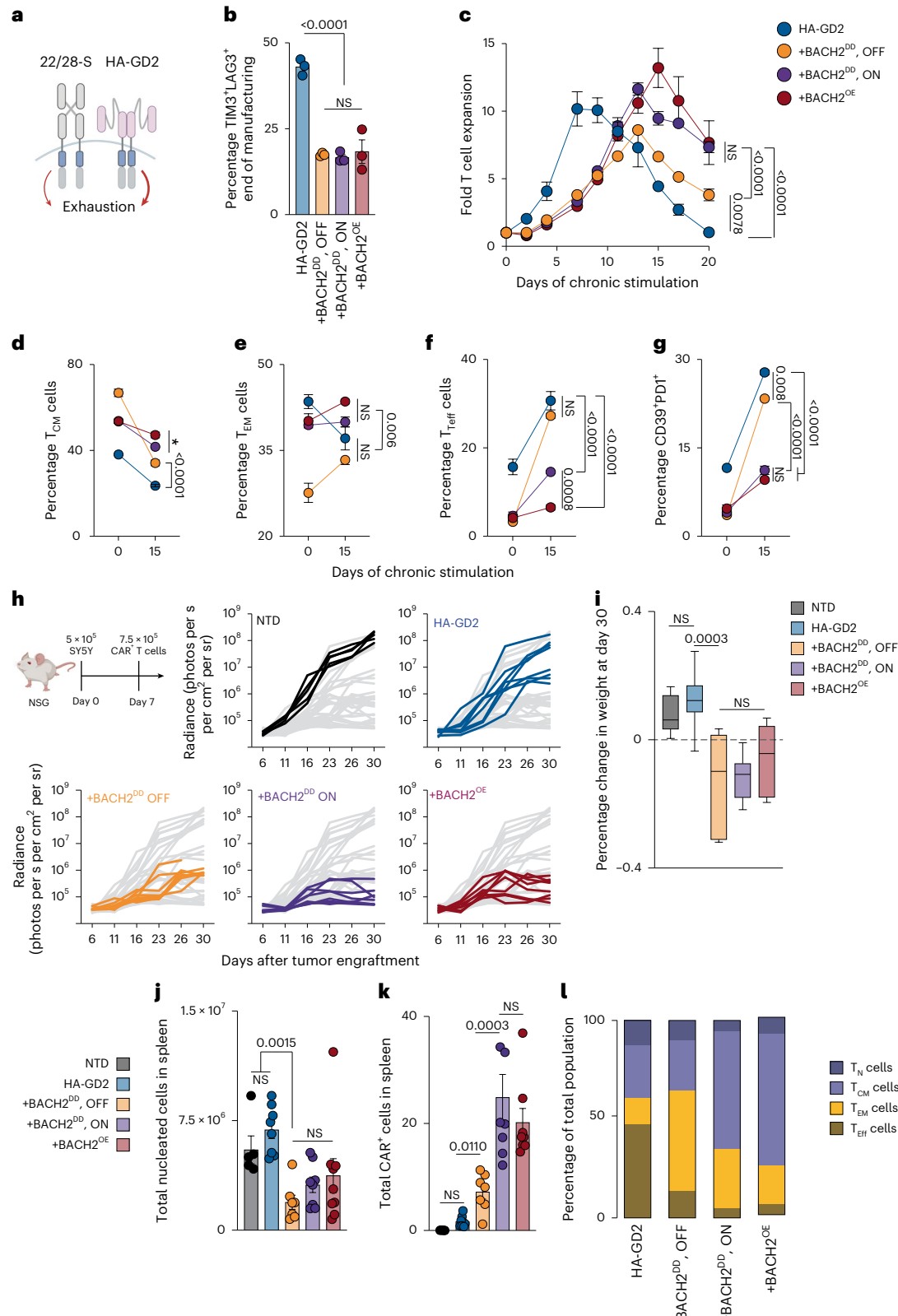

**Fig. 6 | HA-GD2 CAR T cells require higher BACH2 quantity to overcome exhaustion. a**, Schematic of tonic signal strength induced by CD22 or HA-GD2 CARs. **b**, Co-expression of TIM3 and LAG3 by CAR T cells at the end of manufacture. **c**, CAR T cell expansion over time during chronic stimulation with the GD2+ neuroblastoma cell line SY5Y. **d–f**, Central memory (**d**), effector memory (**e**) and terminal effector (**f**) composition of CAR T cells over time during chronic stimulation. **g**, Co-expression of CD39 and PD1 in CAR T cells over time. Representative data are from $n = 3$ donors. **h**, Control of in vivo neuroblastoma. **i**, Percentage change in weight of mice over time after

treatment with CAR T cells 30 d after engraftment. Boxes represent minima and maxima with the line at the mean. **j–l**, Total nucleated cells in the spleen (**j**), fraction of CAR+ T cells in the spleen (**k**) and memory composition of CAR+ T cells in the spleen (**l**). For **h–l**, there are $n = 8$ mice per group except for the NTD control ($n = 5$). All data are presented as either individual values or mean values ± s.e.m. *$P < 0.05$, $P < 0.01$, $P < 0.001$, $P < 0.0001$ by two-way ANOVA with Bonferroni's adjustment for multiple comparisons. NSG, NOD-SCID-γc−/− mice. Schematic in **a** created with BioRender.com.

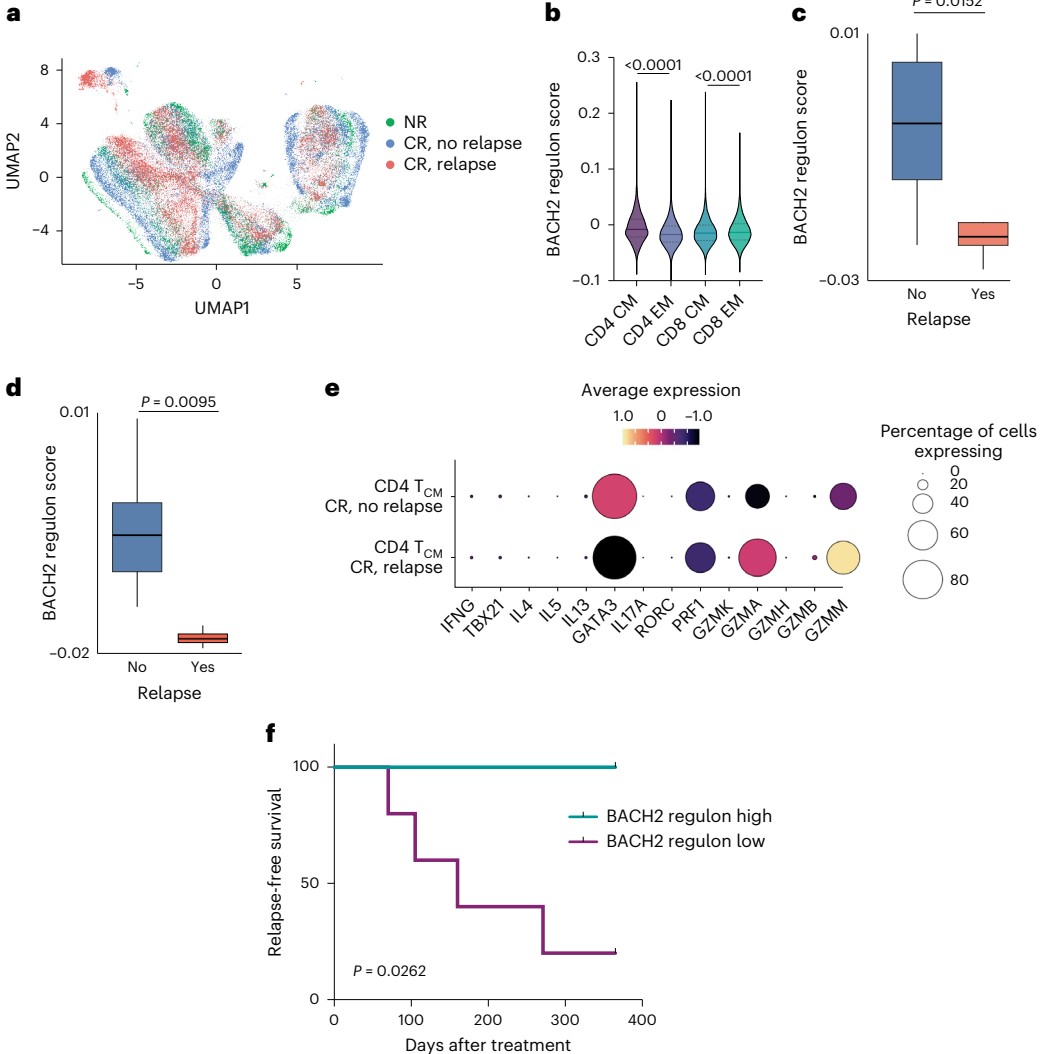

**Fig. 7 | Long-term clinical efficacy of CD19 CAR T cell products correlates with product BACH2 activity. a**, Uniform manifold approximation and projection (UMAP) of CAR T cells from 12 CD19-targeted, 41BB-based, CAR T cell products, labeled by clinical response. **b**, BACH2 regulon score in CD4 and CD8 $T_{CM}$ cells and $T_{EM}$ cells from manufactured CAR T cell products. The analysis was by two-way ANOVA. **c,d**, BACH2 regulon score in bulk (**c**) and CD4 central memory (**d**) from patients who had an initial response and then either relapsed ($n = 3$) or remained in remission ($n = 6$). Boxes represent minima and maxima with the line at mean using significance testing with Student's $t$-test. **e**, Analysis of CD4 $T_{CM}$ cells from patients who relapsed and those who did not. **f**, Relapse-free survival in patients receiving products with either high or low BACH2 regulon scores. The analysis was by log(rank test). CR, complete remission; NR, no response.

observations that BACH2 promotes $T_{CM}$ cell lineage. Of the 9 out of 12 patients who achieved complete remission, 3 of 9 eventually relapsed (33%), reflecting the broad clinical trends for pediatric ALL[34,35]. We found that CAR T cell products given to patients with sustained CRs had higher BACH2 regulon scores (Fig. 7c). This association appeared to be driven primarily by CD4 $T_{CM}$ cells, which had a strong correlation between BACH2 activity and relapse (Fig. 7d) and, to a lesser extent, CD8 $T_{CM}$ cells (Extended Data Fig. 9b). Interrogation of the features defining these CD4 $T_{CM}$ cells revealed high expression of *GATA3* but low expression of cytotoxicity genes (*PRF1*, granzymes) suggesting that these cells were less cytotoxic and more helper 2 T cell-like ($T_H2$ cell-like) than CD4 $T_{CM}$ cells from patients who relapsed (Fig. 7e). We rank-ordered BACH2 regulon scores from each product and designated them as either high or low (above or below mean BACH2 regulon score; Extended Data Fig. 9c) and found that patients receiving products with higher endogenous BACH2 activity were significantly more likely to remain in remission without disease relapse (Fig. 7f). Taken together with our experimental findings that 22/BB-S and 22/28-S⁺BACH2^DD OFF had enhanced anti-leukemic function and that clinical products with tonic 41BB signaling have enhanced efficacy[6,7], these data suggest that

BACH2 activity in CAR T cell products is a core regulator of long-term clinical efficacy.

## Discussion

We demonstrate the broad and dose-dependent impact that the transcriptional regulator BACH2 has on CAR T cell differentiation and functional efficacy against liquid and solid cancers. We found that tonic signaling of CD22 CAR T cells bearing the 41BB domain, which entered clinical trials[36], selectively activated BACH2 during product manufacture and that this endogenous BACH2 promoted stem-like and memory-like programs. Transgenic expression of BACH2 in T cells with dysfunction-inducing CARs enhanced short-term efficacy by preventing exhaustion during manufacture, but restrained long-term function by impairing necessary cell-state transitions. Tuning down the expression of transgenic BACH2 balanced preventing exhaustion and permitting effector function. In a more potent model of exhaustion-inducing tonic CAR signaling, we found that higher levels of BACH2 were needed to prevent exhaustion. Finally, we identified that higher expression of BACH2 in clinical CAR T cell products was associated with improved long-term outcomes in patients with

leukemia, corroborating our findings that low-level BACH2 activity enhances CAR T cell function.

As has been reported previously[12,13], we observed that 41BB signaling induced expression of memory-associated genes and proteins; however, the core mediators of this 41BB-driven memory program remain unknown. Our phenotypic studies demonstrated that BACH2[OE] induced a robust memory and 41BB-like phenotype in 22/28-S cells and transcriptional profiling revealed that 22/28-S⁺BACH2[OE] cells expressed higher levels of memory genes than 22/BB-S. These findings led us to speculate that BACH2 is a core architect of 41BB-driven memory programs. Additional studies are needed to articulate where BACH2 fits in the 41BB-activated regulatory network. Regardless, our findings underscore the complex interplay of memory, effector and exhaustion lineages and identify BACH2 as a central regulator of this rheostat. Many previous studies have identified that the transcriptional programs responsible for transit from resting to activated states can, when dysregulated or imbalanced, push cells into exhausted states[3,5,15,17]. Overcoming exhaustion was critical to enhancing function in both CD22 and GD2 models; although requiring distinct quantities, manipulation of BACH2 may be a broad strategy to tune the exhaustion–effector axis that is central to long-term T cell efficacy. Although the use of a degron that is sensitive to a widely used antibiotic makes this feasible and potentially translatable for clinical evaluation, we are evaluating other degrons that are less likely to invoke the immunogenicity that may result from an *E. coli*-derived protein.

Our degron studies provide key insights into the kinetics of BACH2's impact on T cell functional potential. BACH2[DD] ON and BACH2[OE] cells, in which BACH2 quantity is equivalent during manufacture but differs during antigen engagement, had largely equivalent function in both CD22 and GD2 models. In contrast, BACH2[DD] ON and OFF, in which BACH2 levels differed during manufacture but were the same during antigen engagement, induce fundamentally different T cell function. These comparisons suggest that the primary regulatory impact of BACH2 occurs during product manufacture. Genomic and epigenomic studies are necessary to determine the mechanism of this apparent lineage imprinting, as are studies that develop an engineering approach to further tune BACH2 in a way that prevents exhaustion and permits effector differentiation in more potent models of exhaustion (such as HA-GD2).

We observed that transgenic BACH2 expression, in addition to a profound skewing toward central memory lineage, promoted a slight skewing of CAR products toward CD4 lineages. We also observed that the cell type in infusion products most associated with long-term clinical efficacy was CD4 $T_{CM}$ cells. These observations are reminiscent of a recent study identifying that CAR T cells that persist in patients with >10 years of durable remission are also CD4s[37]. Although those cells were found to express high levels of *GZMK* and *GZMA*, which the CD4 $T_{CM}$ cells in our in vitro studies and clinical products did not, these studies both highlight a central role for CD4 CAR T cells. If this role is classic CD8 help, or nonclassic CD4-driven cytotoxicity, remains to be understood.

We are very intrigued by the unexpected weight loss in mice receiving highly effective GD2 CAR T cells. Although this may have been subclinical GVHD, this seems highly unusual given the lack of changes in fur or skin, lethargy, dehydration or diarrhea and the lack of any such syndrome in our CD22 studies. We are not aware of any other reports of isolated anorexia as a toxicity from immune therapies; however, a previous report demonstrated that a high-affinity GD2 CAR caused neurotoxicity, which, in some cases, manifested as wasting[38]. We are actively pursuing studies to understand the mechanism of this toxicity, including evaluation of off-tumor neurotoxicity that may be nonspecific or GD2 dependent.

The BACH2 functionality identified here mirrors that recently reported for the transcriptional regulator FOXO1 (refs. 39,40). Overexpression of FOXO1 was similarly shown to promote CAR T cell stemness and enhance functionality, whereas expression of a constitutively active FOXO1 limited function. Together with our dose titration studies, these highlight that transcription factor quantity, and not simply presence or absence, is critically important in directing T cell lineage decisions. Like BACH2, FOXO1 also simultaneously prevented exhaustion while compelling memory-like states, further establishing the notion that transcriptional antagonism of exhaustion intrinsically promotes memory. Whether FOXO1 and BACH2 act in parallel to achieve the same developmental outcomes or cooperate in a shared regulatory axis is the focus of ongoing studies.

These findings in FOXO1 and BACH2 also complement studies highlighting the critical balance of memory and effector cells in mediating CAR T cell efficacy. Many studies have demonstrated that products with larger stem and memory compartments lead to enhanced efficacy[27,28,41–43], leading to myriad clinical efforts to enrich these cells in CAR products. In a clear demonstration of a classic immunological principle, and adding important context to these studies, disruption of the mediator complex was shown to enrich effector CAR T cells and drive enhanced tumor control[44]. Consistently, our findings demonstrate that $T_{CM}$ cells enriched in 22/BB-S, 22/28-S⁺BACH2[DD] ON, OFF and 22/28-S⁺BACH2[OE], but only cells that can transit to $T_{EM}$ cells had enhanced efficacy. Collectively, these studies demonstrate that the beneficial impact of early lineage cells is not intrinsic but in their functional potential to successively yield highly active effectors.

## Online content

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

## Methods

### General cell culture, flow cytometry and mass cytometry

T cells and cancer cell lines were grown and cultured at a concentration of $1 \times 10^6$ cells $ml^{-1}$ of standard R10 culture medium (Roswell Park Memorial Institute (RPMI) 1640 + 10% fetal bovine serum, 1% penicillin–streptomycin, 1% Hepes and 1% glutamine) at 37 °C in 5% ambient $CO_2$. For flow cytometry, samples were stained with antibodies against a panel of antigens (Supplementary Table 2) in 100 µl. Samples were acquired using a CyTek Northern Lights spectral flow cytometer and analyzed using FlowJo v9.0 or v10.0 or OMIQ (see Extended Data Fig. 10 for flow cytometry gating strategy).

Mass cytometry was performed as previously described[13]. Briefly, isolated CAR+ T cells were live or dead stained with a short pulse of cisplatin and surface stained for 30 min at room temperature. Cells were then washed and fixed overnight at 4 °C with fix–perm buffer (eBiosciences). Intracellular staining was performed the next day at 4 °C for 1 h. Cells were barcoded according to the manufacturer's instructions (Fluidigm). Cells were washed and suspended in phosphate-buffered saline (PBS) containing 2% paraformaldehyde with Cell-ID Intercalator-IR. Mass cytometry data were collected on a Helios mass cytometer and analyzed using Cytobank (Beckman Coulter) or OMIQ.

For CRISPR editing, a single guide (sg)RNA targeting *BACH2* (GGACTCATACACATACATGG) was designed and purchased from IDT. Ribonucleoprotein (RNP) complexes were formed by incubating 10 µg of TrueCut Cas9 Protein v2 (Invitrogen, cat. no. A36499) with 20 µg of sgRNA for 10 min at room temperature. Resting T cells were washed once with room temperature PBS (Gibco, cat. no. 14190136) and spun at 200*g* for 10 min and resuspended at a concentration of $(2–10) \times 10^6$ cell per 100 µl in Lonza buffer P3 or supplement (Lonza, cat. no. V4XP-3024). The RNP complex and 100 µl of resuspended cells were combined and electroporated using pulse code EO-115 on the Lonza 4D-Nucleofector Core or X Unit.

Cytokine production was quantified using the BD $T_H1$, $T_H2$, $T_H17$ cytokine CBA (BD, cat. no. 560484). Briefly, 48 h after establishing co-cultures, 75 µl of medium supernatant was collected and flash frozen. Supernatants were thawed and cytokines quantified as per the manufacturer's instructions.

### Human T cell engineering and manufacture

Lentiviral vectors were manufactured as previously described[13]. Briefly, pMDG.1 or pCocalEnv (7 µg), pRSV.rev (18 µg), pMDLg/p.RRE (18 µg) packaging plasmids and 15 µg of expression plasmids were mixed and transfected into 293T cells using Lipofectamine 3000 (Invitrogen, cat. no. L3000150) according to the manufacturer's protocol. At both 24 h and 48 h after transfection, the supernatant was collected and filtered through 0.45-µm aPES filters (Thermo Fisher Scientific, cat. no. 1650045). Virus-containing medium was then concentrated using high-speed centrifugation (8,500*g*, 16–18 h at 4 °C with attenuated deceleration). Virus particles were resuspended in 293T growth medium (1:100 to 1:200 of the original volume) and snap-frozen. The virus was stored at −80 °C before usage. For T cell engineering, peripheral blood mononuclear cells were procured from leukoreduction chambers; CD4 and CD8 cells were purified using magnetic beads (Miltenyi Biotec) and combined at a 1:1 ratio before freezing. T cells were activated using CD3 or CD28 stimulatory beads (DynaBeads Thermo Fisher Scientific, cat. no. 40203D) at a ratio of three beads per cell and incubated at 37 °C. Then, 24–30 h after stimulation, lentiviral vectors were added to stimulatory T cell or bead cultures at a multiplicity of infection of 2–4. Beads were removed after 4–6 d of stimulation. Engineered cells were cultured and expanded in R10 and replenished every 2–3 d until freezing at days 14–16. Cells were purified for lentiviral transduction using magnetic beads: cells expressing only CAR constructs were purified using anti-CD34 phycoerythrin (PE) antibody (Beckman, cat. no. IM1459U), followed by anti-PE beads (Miltenyi, cat. no. 130-048-801), whereas cells expressing CAR and

BACH2 constructs were purified using anti-EGFR beads (Miltenyi, cat. no. 130-110-528, same anti-PE bead). This strategy ensured that, in functional assays, all CAR+ cells were also BACH2+. Before functional studies, the number of CAR+ T cells was normalized in each culture. For the $BACH2^{DD}$ ON conditions, manufacturing cultures included 0.1 µM TMP to stabilize BACH2 protein. CD22 CAR constructs all used the m971 scFv and contained the CD8α hinge domain. For CD28-based CARs, the transmembrane was derived from CD28; for 41BB-based CARs the hinge domain was derived from CD8α. All CARs contained the CD3ζ signaling domain. GD2-targeted CARs contained the HA-GD2 scFv[3,4] followed by the CD8α hinge, CD28 transmembrane and CD28 and CD3ζ signaling domains. CD19 CAR studies used the FMC63 scFv with the same structure as 22/28-L.

### In vitro co-culture assays

For short-term co-culture assays, GFP+ Nalm6 cells (originally obtained from American Type Culture Collection (ATCC), cat. no. CRL-3273 and then engineered with GFP lentiviral vectors) were combined with T cells at E-to-T ratios of 1:2 and co-cultures were evaluated for absolute count of target and T cells by flow cytometry. All co-cultures were established in technical triplicate. Cancer cell survival is measured over time but no additional measurements of the E-to-T ratio are made for short-term assays.

Chronic stimulation assays were performed as previously described[13,22]. Briefly, for suspension cultures, the absolute count of manufactured CD34+ T cells and GFP+ Nalm6 was evaluated by flow cytometry and combined with an E-to-T ratio of 1:4. The absolute count of T cell and Nalm6 in this co-culture was re-evaluated every 2–3 d by flow cytometry and fresh cancer cells were added to maintain an E-to-T ratio of 1:4 until T cell failure (reduction in T cell expansion). Similar conditions were established for SY5Y (also originally obtained from ATCC, cat. no. CRL-2266, and engineered with GFP lentiviral vector); however, cancer cell counts were not measured after the first time point for this adherent cell line. T cells were serially replated with fresh SY5Y cells at an E-to-T ratio of 1:4 until T cell contraction.

### RNA-seq and analysis

Total RNA was extracted from engineered T cells at the conclusion of manufacture using QIAzol (QIAGEN) and recovered by RNA Clean and Concentrator spin columns (Zymo). Samples were prepared according to library kit manufacturer's protocol, indexed, pooled and sequenced on an Illumina NovaSeq 6000. Basecalls and demultiplexing were performed with Illumina's bcl2fastq2 software. RNA-seq reads were then aligned and quantified to the Ensembl release 101 primary assembly with an Illumina DRAGEN Bio-IT on-premise server running v3.9.3-8 software. All gene counts were then imported into the R or Bioconductor package EdgeR and trimmed mean of M-values normalization size factors were calculated to adjust for samples for differences in library size. The trimmed mean of M-values size factors and the matrix of counts were then imported into the R or Bioconductor package Limma. Weighted likelihoods based on the observed mean-variance relationship of every gene and sample were then calculated for all samples and the count matrix was transformed to moderated $\log_2$(counts per million) with Limma's voomWithQualityWeights. Differential expression analysis was then performed to analyze for differences between conditions and the results were filtered for only those genes with Benjamini–Hochberg false discovery rate-adjusted $P \le 0.05$. GSEA was done using GSEA v4.1.0.

### ATAC–seq and CUT&RUN

ATAC library preparation was done using standard methods[45]; 500,000 CAR T cells were used per reaction and index primers for sequencing acquired were from IDT for Illumina DNA or RNA UD Indexes plate A (Illumina, cat. no. 20027213). Then, size selection of fragments (<600 bp) was done using SPRIselect Beads (Beckman Coulter, Inc.).

CUT&RUN sample preparation was performed using the CUTANA CUT&RUN kit (EpiCypher, cat. no. 14-1048). Briefly, 500,000 nuclei per reaction were extracted from thawed CAR T cells using nuclei extraction buffer (EpiCypher, cat. no. 21-1026). Then, 1 µg of antibody (control immunoglobulin G, H3K4Me3 or cJUN) was used to pull down fragments. Sequencing libraries were prepared using the CUTANA Library Prep Kit (Epicypher, cat. nos. 14-1001 and 14-1002) and following the manufacturer's instructions.

CUT&RUN and ATAC–seq were analyzed as previously described[46]. In brief, paired-end reads were trimmed for adapters and low-quality bases and aligned to hg38 using Bowtie v2.5.4 (ref. [47]). PCR duplicates were removed and for CUT&RUN fragments were truncated to 120 bp to enrich mononucleosome-sized fragments. Sorted BAM files were used for peak calling with MACS2 (ref. [48]) and the resulting narrow peaks were filtered against the ENCODE blacklist. Peak sequences were extracted for downstream motif and footprinting analyses. Differential binding analysis was performed using DiffBind (v3.18.0) with a DESeq2 (1.48.2)[49] framework and default settings after resizing peaks to 500 bp around summits. Genomic annotation of peaks and nearest-gene mapping were performed using ChIPseeker (v1.44.0)[50] with the default settings.

## Confocal microscopy

Cells were cultured in RPMI and washed once with cold PBS before initiating staining. The cells were blocked with 1% bovine serum albumin for 15 min and then stained with CD34-APC antibody (BD Pharmingen, cat. no. 555824), a marker for CAR+ cells, for 40 min at room temperature. Subsequently, the stained cells were deposited on to glass slides using cytospin centrifugation. Furthermore, samples were fixed and permeabilized at room temperature using 4% paraformaldehyde for 20 min followed by 0.1% Triton X-100 in PBS for 15 min. Intracellular staining was performed using the BACH2-PE antibody (BioLegend cat. no. 695603) for 1 h at room temperature, followed by DAPI staining (BD Pharmingen, cat. no.564907) for 5 min at room temperature. Stained cells were mounted in ProLong antifade (Invitrogen, cat. no. P36980) and imaged on the Nikon AX-R confocal microscope with a Plan-Apochromat λD ×100 oil OFN25 DIC N2 1.4 5 numerical aperture objective lens and multi-band dichroic beamsplitter set for 405-nm, 488-nm, 561-nm and 640-nm laser excitation. DAPI, R-phycoerythrin (R-PE) and allophycocyanin were excited with 405-nm, 561-nm and 640-nm lasers, respectively, and detected using gallium arsenide phosphide cathode detectors. For analysis, three-dimensional surfaces were generated either automatically (for DAPI) or manually (for allophycocyanin and R-PE) by drawing contours at a z-interval of 8–10 slices per z-stack images, which comprise 20–25 slices at 0.2-µm step size in Imaris (v10.1.1). Masks were then created from the surfaces for individual channels for subsequent image analysis where the intensity values for R-PE were extracted from the masked nuclei and whole cells.

## In vivo studies

The 6–10-week-old NOD-SCID-γc$^{-/-}$ mice were obtained from the Jackson Laboratory and maintained in pathogen-free conditions. Animals were injected via the tail vein with either $1 \times 10^6$ Nalm6 cells or $5 \times 10^5$ SY5Y cells in 0.2 ml of sterile PBS. For ALL studies, $2.0 \times 10^6$ CAR+ T cells were injected via the tail vein in 0.2 ml of sterile PBS on day 10 after tumor injection; for neuroblastoma studies $7.5 \times 10^5$ CAR+ T cells were delivered via the tail vein. Animals were randomized to ensure equivalent disease burden in each treatment arm. For in vivo CAR T cell evaluation, animals were engrafted with disease and given CAR T cells as described and then sacrificed at days 7 and 13. Spleens were collected, dissociated, made into single-cell suspensions, counted and then evaluated by flow cytometry. Animals were monitored for signs of disease progression and overt toxicity, such as xenogeneic GVHD, as evidenced by >15% loss in body weight, loss of fur, diarrhea, conjunctivitis and disease-related hind-limb paralysis. Disease burdens were monitored over time using a Spectral Instruments Imaging AMI Instrument with analysis done using associated Aura software. Animals were sacrificed when radiance reached >$3 \times 10^9$ photos per s per cm$^2$ per sr (5 log above background). To avoid skewing of radiance data, graphic representation for each group was stopped after the death of the first animal in the group. All studies were performed under the approval of the Washington University Institutional Animal Care and Use Committee (protocol no. 25-0028). No statistical methods were used to predetermine sample sizes but we used sizes consistent with many previous studies. Data collection for animal studies was blinded. No animals were excluded from analysis.

## scRNA-seq

scRNA-seq data for CAR T cells were extracted from a publicly available dataset that has been previously published[47] and associated clinical data[51]. Only infusion products were analyzed. Cell annotation were extracted from the published Seurat object, which applied the SingleR algorithm[32]. After paring down to CD4+ and CD8+ subsets, we also excluded cells from patients lacking sufficient clinical information (for example, missing relapse status or response data). For all relapse-based comparisons, we retained only the cells from patients with relapse statuses coded as 'Yes' or 'No', removing any nonresponder entries for the purposes of this analysis. Next, to investigate transcriptional activity associated with BACH2, we computed a BACH2 regulon expression score for each cell, using a curated list of BACH2 target genes created using the SCENIC package[33]. The BACH2 regulon score was calculated as the average expression of BACH2 target genes relative to a set of control features using the AddModuleScore feature of the Seurat package. To avoid potential biases at the single-cell level for certain visualization and statistical comparisons, we aggregated these values by patient: for each patient, we calculated the mean BACH2 module score across the relevant T cell subsets. This patient-level aggregation helped ensure that each patient served as a single biological replicate in subsequent statistical tests. For visualizations, we primarily used box plots to compare BACH2 module scores across categories such as relapse status, response group and T cell subtype. In addition, we generated Uniform Manifold Approximation and Projection embeddings of the single-cell data for qualitative assessment of cluster structures. Kaplan–Meier survival analyses were performed using the 'survival' and 'survminer' R packages, with time to relapse as the endpoint and BACH2 module scores dichotomized at their median into 'high' versus 'low' groups.

## Statistical analysis

All comparisons between two groups were performed using a two-tailed, unpaired Student's t-test. Comparisons between more than two groups were performed by two-way ANOVA with Bonferroni's correction for multiple comparisons. Survival data were analyzed using the log(rank (Mantel–Cox) test. All results are represented as mean ± s.e.m.

## Reporting summary

Further information on research design is available in the Nature Portfolio Reporting Summary linked to this article.

## Data availability

All bulk RNA-seq, ATAC–seq and CUT&RUN data have been deposited in the Gene Expression Omnibus under accession no. GSE287286. Requests for scRNA-seq from clinical CAR T cell products can be directed to S.G. All other data and materials will be made available upon request to the corresponding author. Source data are provided with this paper.

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

## Acknowledgements

This work was supported by the Washington University Children's Discovery Institute, Damon Runyon Clinical Scientist Award, Gilead Research Scholars Program and V Foundation for Cancer Research (all to N.S.). We thank K. Murphy, R. Roychoudhuri, T. Wu and C. Yao for helpful discussions. Schematics in Figs. 1a, 3a, 5a and 6a created with BioRender.com.

## Author contributions

T.-C.C., A. Heard, J.L., J.M.W., A.B., J.H.L., J.-F.C., M.E.S., Y.-S.H., H.S., Y.T., V.G., B.T., S.A., D.K.G., A. Horn, J.R. and N.S. performed the research. A.J.F., M.P.M., E.W.W., T.J.W., J.F.D., J.C.C., S.G. and P.G.T. provided critical reagents, technical or computational expertise and guidance. T.-C.C., A. Heard and N.S. designed the research. T.-C.C. and N.S. wrote the paper. All authors reviewed the paper.

## Competing interests

M.E.S. is currently an employee of Wugen. E.W.W. holds equity in Lyell Immunopharma and consults for Umoja Immunopharma. T.J.W. is a founder and consultant for Obsidian Therapeutics. J.F.D. receives research funding from Amphivena Therapeutics, NeoImmuneTech, Macrogenics, Incyte, Bioline Rx and Wugen; has equity ownership in Magenta Therapeutics and Wugen; consults for Incyte, RiverVest Venture Partners and hC Bioscience, Inc.; and is a board member for Magenta Therapeutics. J.C.C. and P.G.T. have patents related to cellular immunotherapy. S.G. is a co-inventor on patents and patent applications in the fields of gene and cell therapy for cancer, a member of the Scientific Advisory Board of Be Biopharma and a Data Safety Monitoring Board member of Immatics and has received honoraria from CARGO Therapeutics within the last year. N.S. holds equity in Phoreus Bio and Defiance Therapeutics and has patents and patent applications in the field of cellular immunotherapy. T.-C.C. and N.S. have submitted patent applications related to this work. The other authors declare no competing interests.

## Additional information

**Extended data** is available for this paper at https://doi.org/10.1038/s41590-025-02391-5.

**Correspondence and requests for materials** should be addressed to Nathan Singh.

**a**

22/28-S
motifs

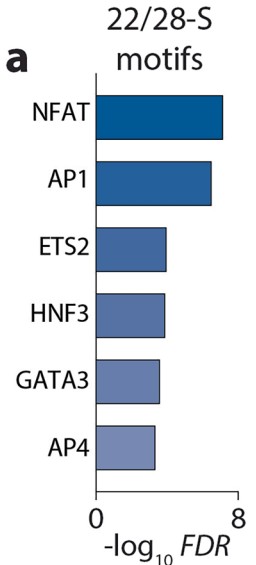

**b**

NFAT-GFP  AP1-mCherry

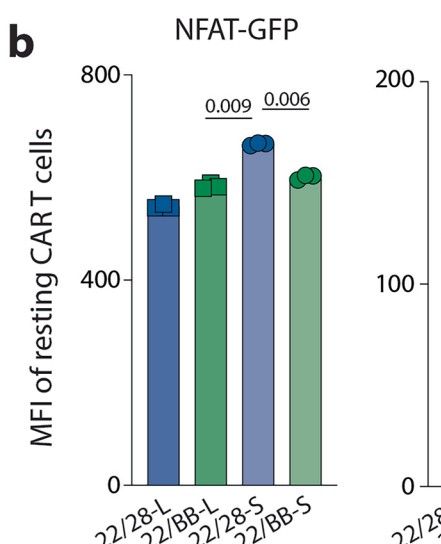

**c**

transcription factor motif analysis
chromVAR

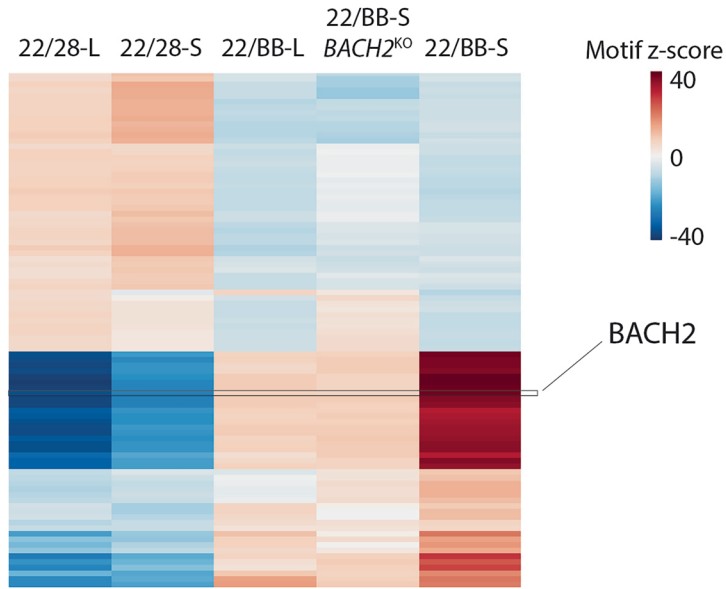

**d**

End of manufacturing

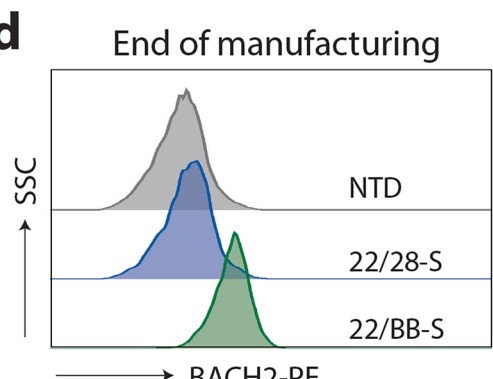

**Extended Data Fig. 1 | Impact of tonic CAR signaling on T cell transcriptional programs. a**, Transcriptional regulators of genes significantly upregulated by 22/28-S, identified by geneset enrichment analysis (GSEA). **b**, Induction of NFAT and AP1 in CAR-expressing Jurkat reporter cells 16 h after co-culture with Nalm6; n = 3 in technical triplicate. **c**, chromVAR analysis of differentially accessible motifs across CAR samples. **d**, Expression of BACH2 at the end of manufacturing in CAR T cells. All data are represented as either individual values or mean values ± SEM. *P < 0.05, **P < 0.01, ***P < 0.001, ****P < 0.0001 by two-way ANOVA with Bonferroni adjustment for multiple comparisons.

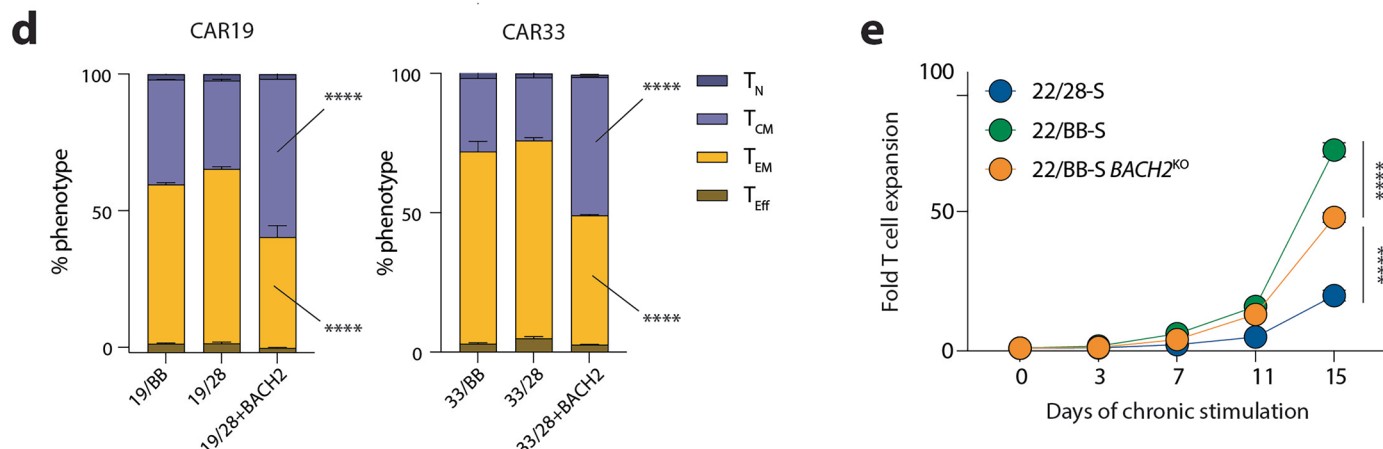

**Extended Data Fig. 2 | See next page for caption.**

**Extended Data Fig. 2 | BACH2 overexpression alters CAR T cell phenotypes.**
**a**, Expression of BACH2 in bulk, CD4 and CD8 CAR T cells at the conclusion of
manufacturing. **b**, Expression of TIM3 and LAG3 in CAR T cells at the end of
manufacturing (n = 3 donors). **c**, Memory composition of bulk, CD4 and CD8 CAR
T cells at the conclusion of manufacturing. **d**, Memory subtype composition of

CD19 and CD33 CAR T cell products (representative of n = 2 donors). **e**, Expansion
of CAR T cells over time in a chronic stimulation assay. All data are represented as
either individual values or mean values ± SEM. *P < 0.05, **P < 0.01, ***P < 0.001,
****P < 0.0001 by two-way ANOVA with Bonferroni adjustment for multiple
comparisons.

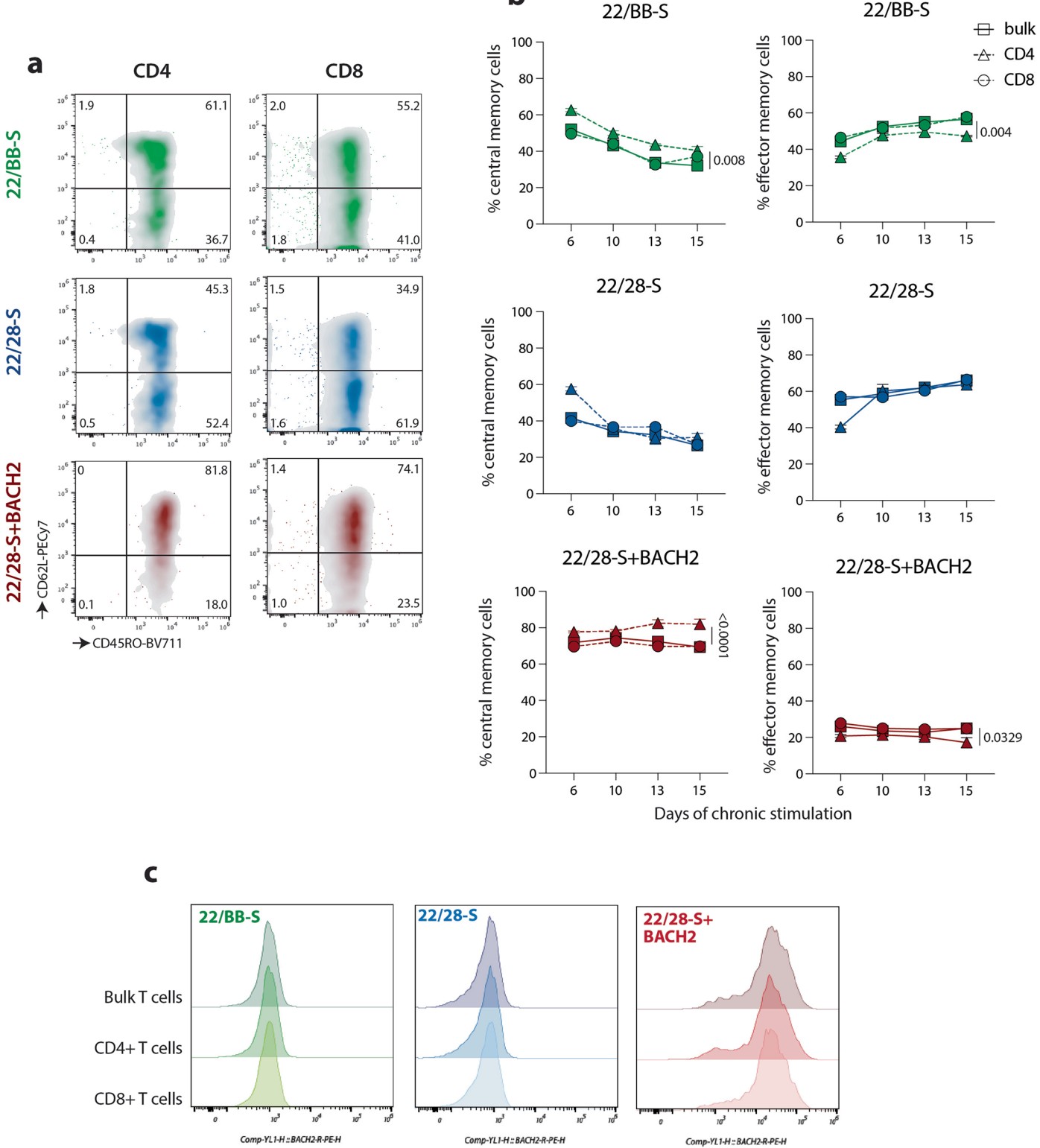

**Extended Data Fig. 3 | BACH2 overexpression alters CAR T cell lineage commitment in response to prolonged antigen stimulation. a**, Representative flow cytometry of memory subtypes at day 6 of chronic stimulation. **b**, Memory subtype composition of bulk, CD4 and CD8 CAR T cells over the course of chronic stimulation. **c**, BACH2 expression in bulk, CD4 and CD8 CAR T cells at the end of chronic stimulation cultures. All data representative of n = 3 donors. All data are represented as either individual values or mean values ± SEM. *P < 0.05, **P < 0.01, ***P < 0.001, ****P < 0.0001 by two-way ANOVA with Bonferroni adjustment for multiple comparisons.

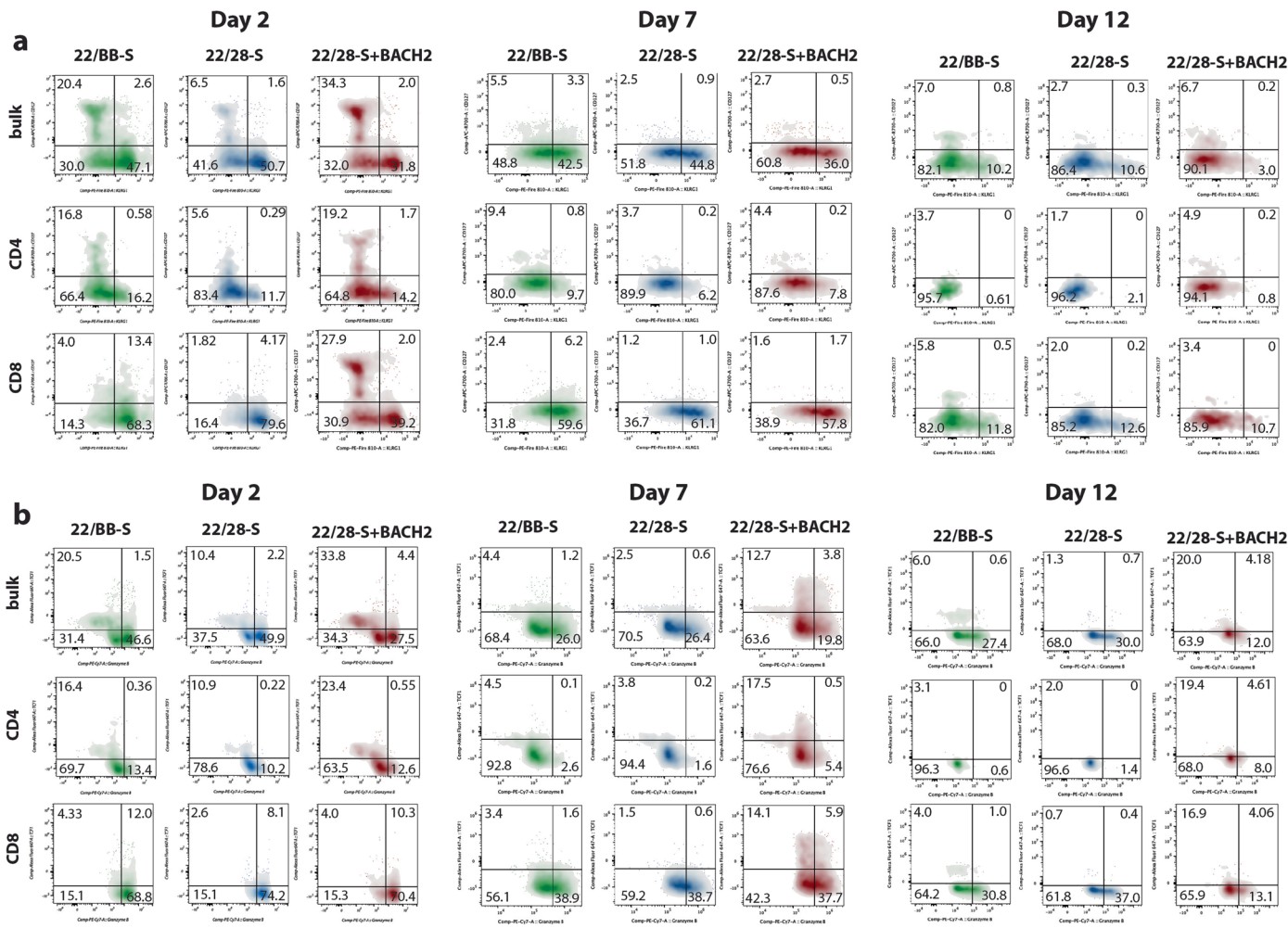

**Extended Data Fig. 4 | BACH2 overexpression skews CAR T cell memory and effector markers. a**, **b**, Representative plots of longitudinal flow cytometry analyzing **a**, CD127/KLRG1 and **b**, TCF1vGZMB expression in bulk, CD4 and CD8 CAR + T cells.

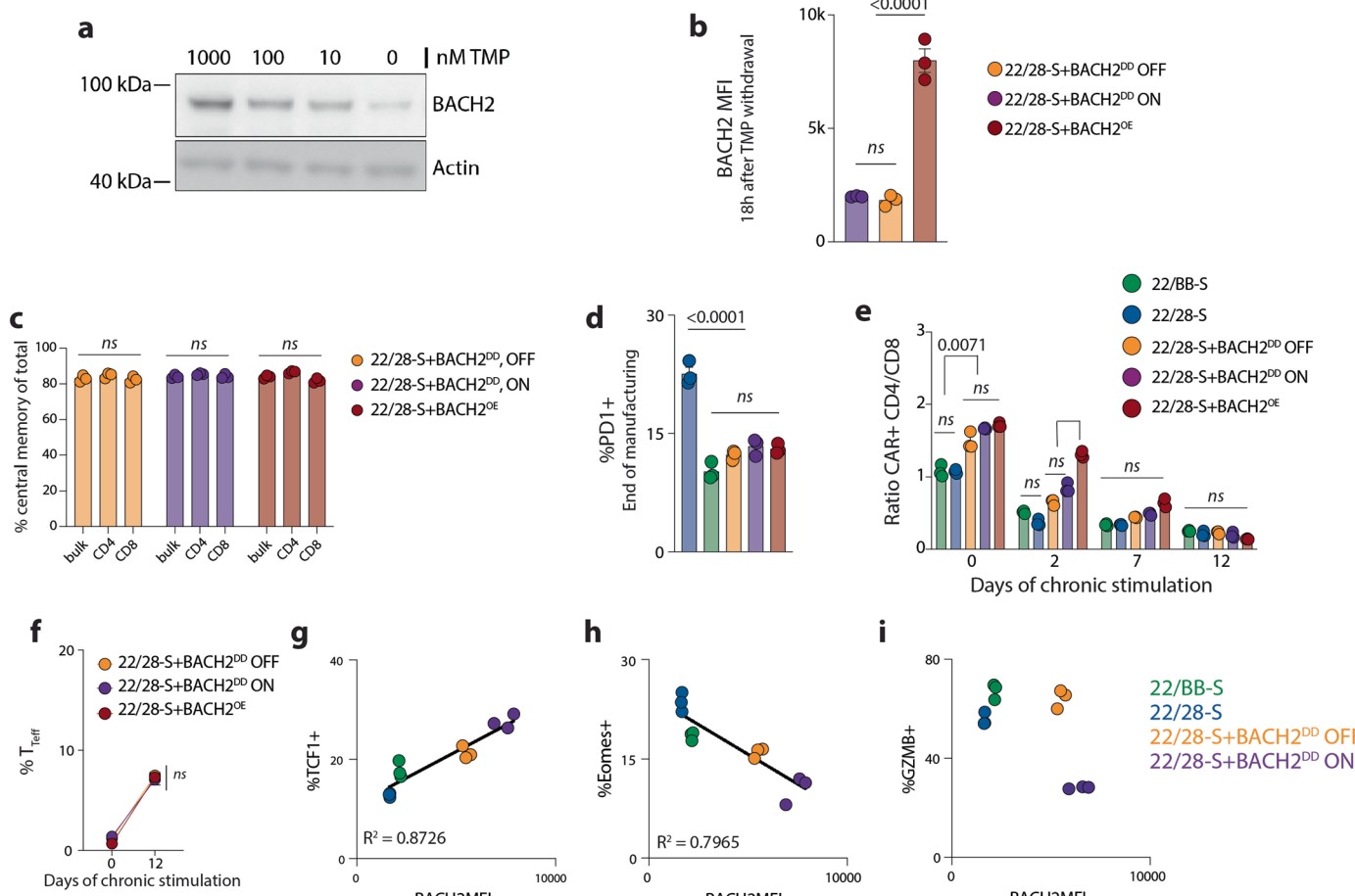

**Extended Data Fig. 5 | Tagging BACH2 with a degron enables control of lineage commitment. a**, Western blot of BACH2 in T cells expressing BACH2$^{DD}$ exposed to different concentrations of TMP. **b**, BACH2 quantity in CAR T cells; TMP was withdrawn from BACH2$^{DD}$ ON CAR T cells culture media for 18 h. **c**, TCM frequency in bulk, CD4 and CD8 CAR T cells at the end of manufacturing. **d**, PD1 expression in CAR T cells at the end of manufacturing. **e**, CD4/CD8 ratio of CAR T cells over time. **f**, Terminal effector T cell composition of CAR T cell products over time. **g–i**, Correlation between BACH2 quantity and **g**, TCF1 **h**, Eomes and **i**, GZMB in CAR T cells. **b–i**, representative of n = 3 donors. **g**, **h**, simple linear regression analysis. All other experiments, *P < 0.05, **P < 0.01, ***P < 0.001, ****P < 0.0001 by two-way ANOVA with Bonferroni adjustment for multiple comparisons. All data are represented as either individual values or mean values ± SEM.

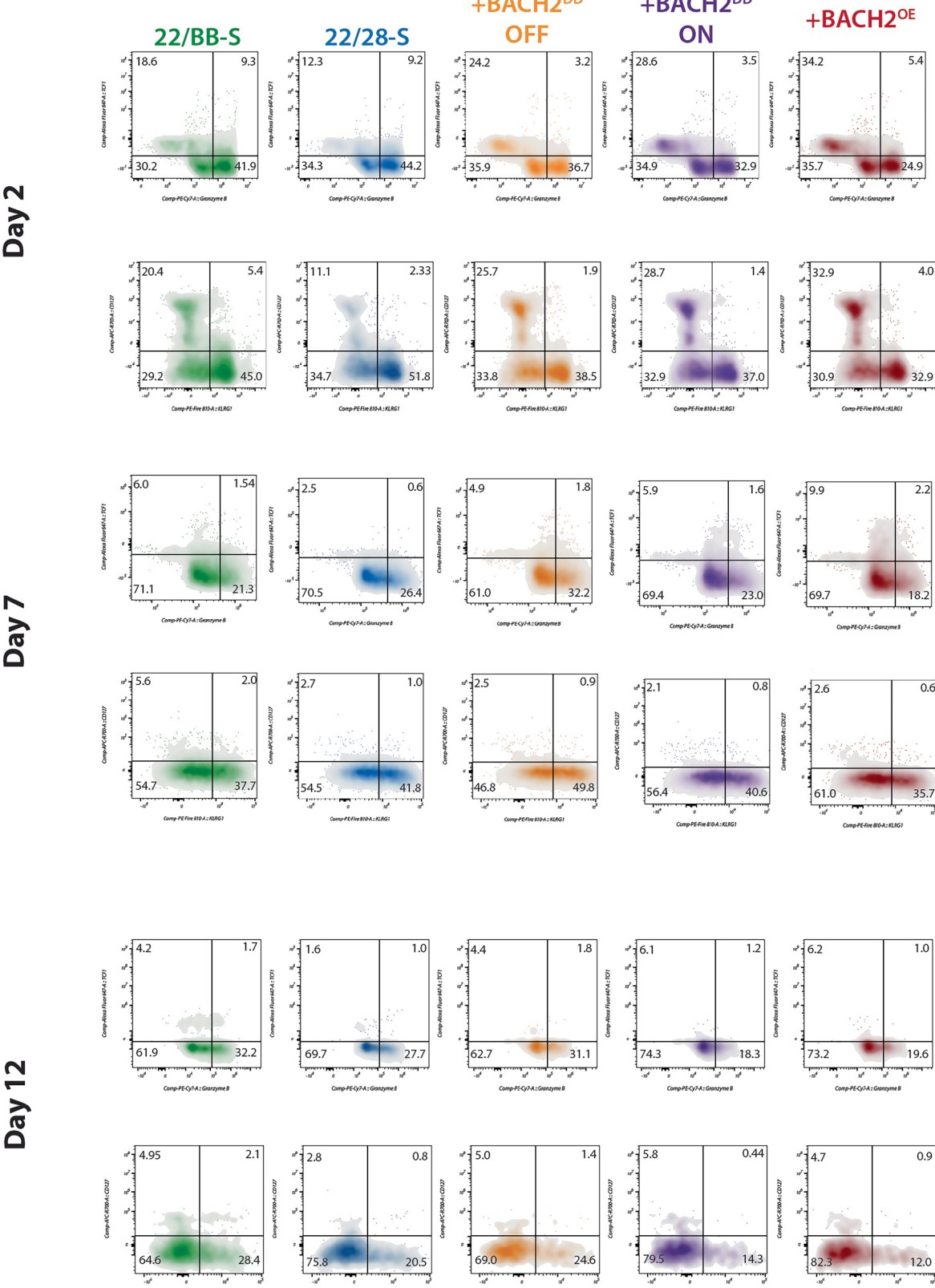

**Extended Data Fig. 6 | CAR T cell lineage commitment in response to chronic antigen stimulation using degron-tagged BACH2.** Representative plots of longitudinal flow cytometry analyzing CD127/KLRG1 and TCF1vGZMB expression.

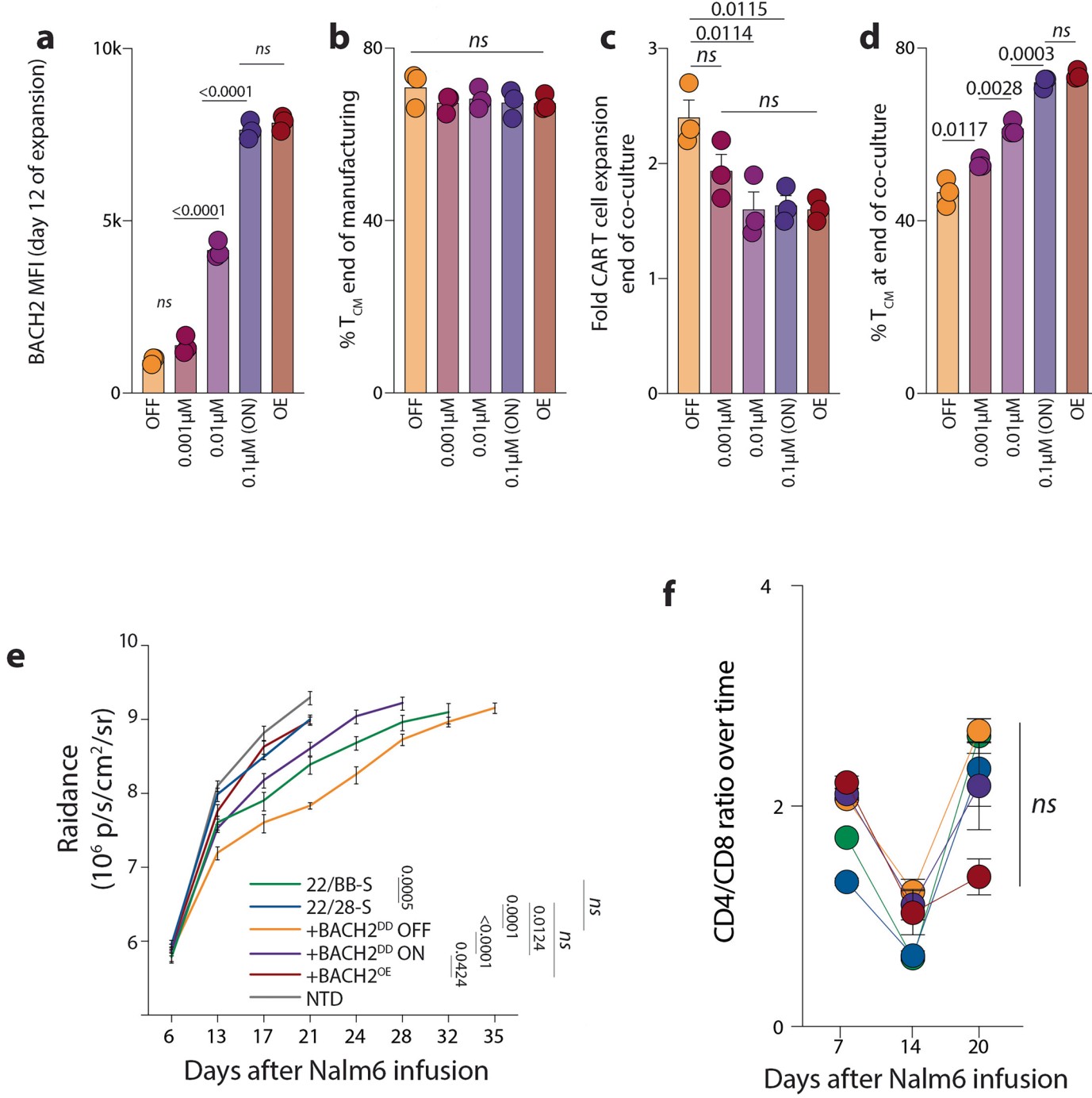

**Extended Data Fig. 7 | Titrated control of CAR T cell function using varied trimethoprim concentrations. a–d,** Evaluation of the impact of TMP dose on **a,** BACH2 MFI during manufacturing, **b,** % of products that were CD62L + CD45RO+ at the end of manufacturing, **c,** fold expansion of CAR T cell at the end of chronic stimulation and **d,** % of products that were CD62L + CD45RO+ at the end of chronic stimulation. **a–d,** n = 1 donor performed in technical triplicate.

**e,** Cumulative tumor bioluminescence in mice over time. **f,** CD4/CD8 ratio of CAR + T cells in mice over time. **e, f,** n = 8 animals per group except NTD control (n = 6). All data are represented as either individual values or mean values ± SEM. *P < 0.05, **P < 0.01, ***P < 0.001, ****P < 0.0001 by two-way ANOVA with Bonferroni adjustment for multiple comparisons.

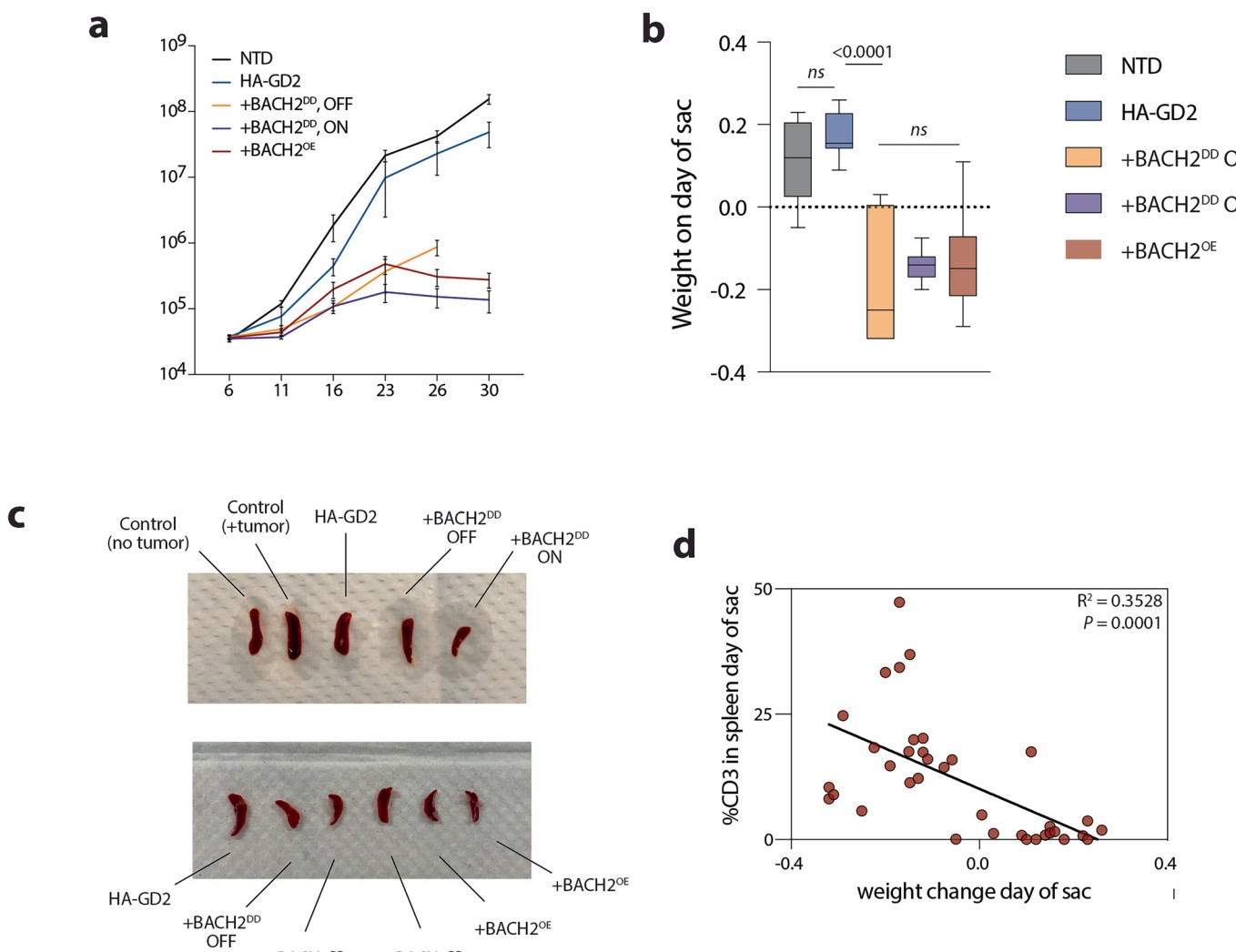

**Extended Data Fig. 8 | BACH2-enhanced GD2 CAR T cells enhance tumor control but cause toxicity in mice. a**, Mean bioluminescent signal from mice engrafted with SY5Y and treated with GD2 CAR T cell products over time. **b**, % change in weight at time of animal sacrifice. Boxes represent minima and maxima with line at mean. **a**, **b**, n = 8 mice per group except NTD control (n = 5).

**c**, Representative photos of mouse spleens at time of sacrifice. **d**, Correlation between % weight loss and % of splenic CAR+ cells done by simple linear regression analysis. All data are represented as either individual values or mean values ± SEM. *P < 0.05, **P < 0.01, ***P < 0.001, ****P < 0.0001 by two-way ANOVA with Bonferroni adjustment for multiple comparisons.

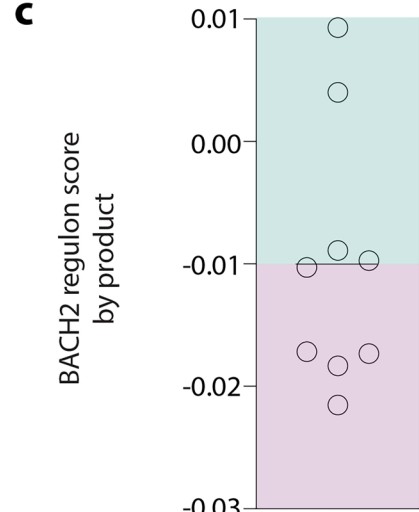

**Extended Data Fig. 9 | Detailed analysis of clinical CAR T cell products.**
**a**, UMAP of CAR T cells colored by product. **b**, BACH2 regulon score in CD8 T central memory, CD4 and CD8 effector memory CAR T cells from patients who had an initial response and then either relapsed (n = 3) or remained in remission (n = 6). Boxes represent minima and maxima with line at mean with significance testing by student's t-test. **c**, Assignment of high or low regulon scores in CAR T cell products.

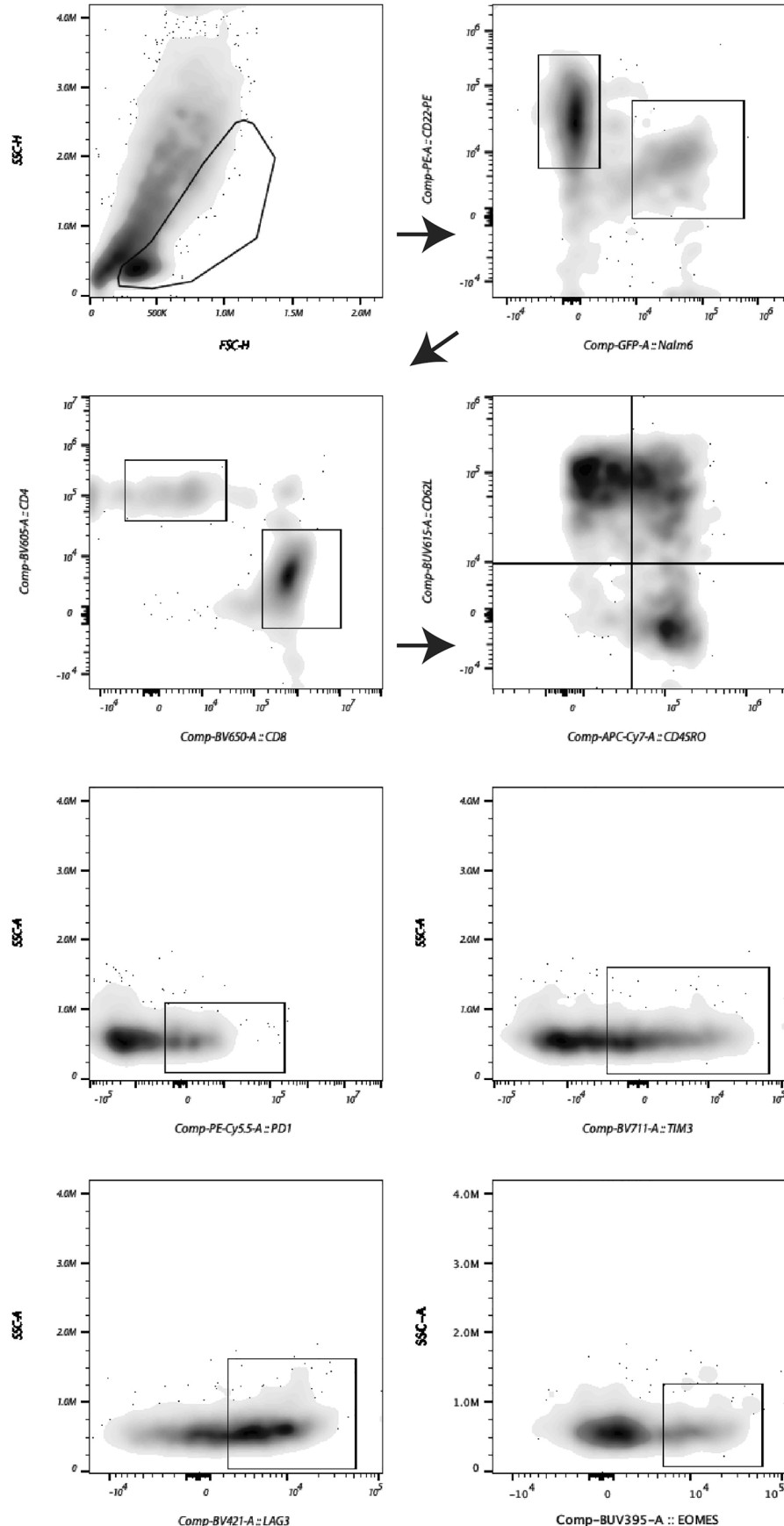

**Extended Data Fig. 10 | Flow cytometry gating strategy.** Schematic of gating strategy for all flow cytometry analyses.

# Reporting Summary

## Statistics

For all statistical analyses, confirm that the following items are present in the figure legend, table legend, main text, or Methods section.

| n/a | Confirmed | |
|---|---|---|
| ☐ | ☒ | The exact sample size (*n*) for each experimental group/condition, given as a discrete number and unit of measurement |
| ☐ | ☒ | A statement on whether measurements were taken from distinct samples or whether the same sample was measured repeatedly |
| ☐ | ☒ | The statistical test(s) used AND whether they are one- or two-sided *Only common tests should be described solely by name; describe more complex techniques in the Methods section.* |
| ☒ | ☐ | A description of all covariates tested |
| ☐ | ☒ | A description of any assumptions or corrections, such as tests of normality and adjustment for multiple comparisons |
| ☐ | ☒ | A full description of the statistical parameters including central tendency (e.g. means) or other basic estimates (e.g. regression coefficient) AND variation (e.g. standard deviation) or associated estimates of uncertainty (e.g. confidence intervals) |
| ☐ | ☒ | For null hypothesis testing, the test statistic (e.g. *F*, *t*, *r*) with confidence intervals, effect sizes, degrees of freedom and *P* value noted *Give P values as exact values whenever suitable.* |
| ☒ | ☐ | For Bayesian analysis, information on the choice of priors and Markov chain Monte Carlo settings |
| ☒ | ☐ | For hierarchical and complex designs, identification of the appropriate level for tests and full reporting of outcomes |
| ☒ | ☐ | Estimates of effect sizes (e.g. Cohen's *d*, Pearson's *r*), indicating how they were calculated |

*Our web collection on statistics for biologists contains articles on many of the points above.*

## Software and code

Policy information about availability of computer code

| Data collection | standard R packages were used to analyze ATACseq, RNAseq, scRNAseq and CUT&RUN data. |
|---|---|
| Data analysis | n/a |

For manuscripts utilizing custom algorithms or software that are central to the research but not yet described in published literature, software must be made available to editors and reviewers. We strongly encourage code deposition in a community repository (e.g. GitHub). See the Nature Portfolio guidelines for submitting code & software for further information.

## Data

Policy information about availability of data

All manuscripts must include a data availability statement. This statement should provide the following information, where applicable:
- Accession codes, unique identifiers, or web links for publicly available datasets
- A description of any restrictions on data availability
- For clinical datasets or third party data, please ensure that the statement adheres to our policy

All bulk RNAseq data have been made publicly available in GEO (GSE287286). ATACseq and CUT&RUN data will be deposited at the same accession site. Requests for scRNAseq from clinical CAR T cell products can be directed to P.G.T. and S.G. All other data and materials will be made available upon request to the corresponding author.

# Research involving human participants, their data, or biological material

Policy information about studies with human participants or human data. See also policy information about sex, gender (identity/presentation), and sexual orientation and race, ethnicity and racism.

| | |
|---|---|
| Reporting on sex and gender | n/a |
| Reporting on race, ethnicity, or other socially relevant groupings | n/a |
| Population characteristics | n/a |
| Recruitment | n/a |
| Ethics oversight | n/a |

Note that full information on the approval of the study protocol must also be provided in the manuscript.

# Field-specific reporting

Please select the one below that is the best fit for your research. If you are not sure, read the appropriate sections before making your selection.

☒ Life sciences　　　☐ Behavioural & social sciences　　　☐ Ecological, evolutionary & environmental sciences

For a reference copy of the document with all sections, see nature.com/documents/nr-reporting-summary-flat.pdf

# Life sciences study design

All studies must disclose on these points even when the disclosure is negative.

| | |
|---|---|
| Sample size | All experiments using human T cells were repeated using 2-5 independent donors and performed in technical triplicate. Studies in mice included 8 animals per condition. Both of these are standard practice for the field. |
| Data exclusions | No data were excluded. |
| Replication | As above, data were repeated using several independent donors. No studies contradicted the findings presented and aggregate findings are presented when possible. |
| Randomization | For in vitro studies, randomization does not apply. For in vivo studies, mice were randomized to treatment groups to ensure equivalent disease burden at time of treatment. |
| Blinding | Animal disease burden measurements were performed in a blinded fashion. |

# Reporting for specific materials, systems and methods

We require information from authors about some types of materials, experimental systems and methods used in many studies. Here, indicate whether each material, system or method listed is relevant to your study. If you are not sure if a list item applies to your research, read the appropriate section before selecting a response.

## Materials & experimental systems

| n/a | Involved in the study |
|---|---|
| ☐ | ☒ Antibodies |
| ☐ | ☒ Eukaryotic cell lines |
| ☒ | ☐ Palaeontology and archaeology |
| ☐ | ☒ Animals and other organisms |
| ☒ | ☐ Clinical data |
| ☒ | ☐ Dual use research of concern |
| ☒ | ☐ Plants |

## Methods

| n/a | Involved in the study |
|---|---|
| ☒ | ☐ ChIP-seq |
| ☐ | ☒ Flow cytometry |
| ☒ | ☐ MRI-based neuroimaging |

# Antibodies

| | |
|---|---|
| Antibodies used | We have included these in Supplementary Tables 1 and 2. |

| Validation | Antibodies were used as per manufacturer instruction and validated by manufacturer website descriptions. |

# Eukaryotic cell lines

Policy information about cell lines and Sex and Gender in Research

| Cell line source(s) | Nalm6, 293T and SY5Y human-derived cell lines were originally obtained from ATCC |
| Authentication | Cell lines are authenticated using STR testing annually. |
| Mycoplasma contamination | All cell line cultures are tested for mycoplasma after 4 weeks in culture. |
| Commonly misidentified lines (See ICLAC register) | n/a |

# Animals and other research organisms

Policy information about studies involving animals; ARRIVE guidelines recommended for reporting animal research, and Sex and Gender in Research

| Laboratory animals | 6-10 week old NOD-SCID-γc-/- (NSG) mice were obtained from the Jackson Laboratory |
| Wild animals | n/a |
| Reporting on sex | All animals were female. |
| Field-collected samples | n/a |
| Ethics oversight | All studies were performed under the approval of the Washington University Institutional Animal Care and Use Committee (IACUC, protocol #22-0028). |

Note that full information on the approval of the study protocol must also be provided in the manuscript.

# Plants

| Seed stocks | n/a |
| Novel plant genotypes | n/a |
| Authentication | n/a |

# Flow Cytometry

## Plots

Confirm that:

☒ The axis labels state the marker and fluorochrome used (e.g. CD4-FITC).

☒ The axis scales are clearly visible. Include numbers along axes only for bottom left plot of group (a 'group' is an analysis of identical markers).

☒ All plots are contour plots with outliers or pseudocolor plots.

☒ A numerical value for number of cells or percentage (with statistics) is provided.

## Methodology

| Sample preparation | samples were stained with antibodies against a panel of antigens (see Supplementary Table 2) in 100μL for 30 minutes at room temperature or 1 hour at 4C. Samples were washed twice with FACS buffer (PBS+2% FBS) and then analyzed. |
| Instrument | CyTek Northern Lights or ThermoFisher Attune |
| Software | FlowJo v9.0 or 10.0 or OMIQ. |

| Cell population abundance | for experiments in which T cells were sorted, purity was always >90% as assessed by independent transduction markers. |
|---|---|
| Gating strategy | Live cell gates were established using FSC and SSC dot plots, followed gating for 7AAD-negative cells. Positive and negative gates were defined using appropriate positive and negative controls. Specific gating approaches varied depending on the panel complexity and antigen abundance. |

☒ Tick this box to confirm that a figure exemplifying the gating strategy is provided in the Supplementary Information.

