## [Peer Review File · Nature Immunology]

BACH2 regulates T cell lineage states to enhance CAR T cell function

Corresponding Author: Dr Nathan Singh

Version 0:

Reviewer comments:

Reviewer #1

(Remarks to the Author)

In this work, the authors study the effect of over-expression of BACH2 on the functionality of T cells being expanded for CAR therapy in vitro, and how this molecule contributes to the anti-tumor killing ability of the CART cells. The authors find that if they force expression of BACH2, they can increase the expansion potential of CD28 tailed CARs, which usually have the problem of crashing after a short time in vitro. The challenge of just over-expressing Bach2 is that this molecule inhibits effector differentiation, so the cells are actually worse in vivo. However, the big highlight of the work is the authors developed a very novel method to allow timed deletion of the Bach2 molecule using an ecoli derived degradation domain that allowed them to turn Bach2 on and off with an antibiotic. Using this approach they take all the benefits of expressing Bach2 during expansion, but then using their novel system, turn it off by withdrawing the antibiotic. Overall, this is a very clever system to allow timed expression of BACH2 to maximize its inhibitory role during expansion, but then remove the effect when the cells need to take on an effector state. The idea is novel and builds on strong previous fundamental findings related to the role of BACH2 in T cell function and certainly contributes to new ideas on how to use CAR T cells in cancer. My biggest issue is that there are quite a lot of places in the current version of the manuscript that need additional analysis of the cells to provide better understanding of how BACH2 is altering the phenotype and function in vitro and in vivo over time and some more detail around experimental setups and interpretation.

1. I expect the authors have a lot of these data, and I don't fault them at all, but throughout the paper there needs to be inclusion of representative FACS plots of all markers examined. Other labs will need to repeat this work and will need to see what the authors are presenting so they can see if the observations are reproducible. All FACS plots need to show outlier dots if they are contours, axis ticks with the scale, and the fluorescent color/metal of the marker used. I have no doubts about the authors claims, but readers need to be able to compare to what they find in their experiments or what is seen in other pieces of work.

2. The major experimental additions that are needed are all related to providing more detailed analysis of the T cells in the different conditions tested. The authors are generally making a lot of claims about T cell exhaustion and other states. They have some excellent functional data like tumor killing or the ability of the cells to expand, but there is very little detail of how the cells respond over time or in vivo.

i) Throughout the paper, the authors need to specify if they are looking at CD4 or CD8 T cells when they are showing phenotypic changes. There are good reasons to think that CD4 and CD8 T cells might not respond in the same way to BACH2 overexpression and the interpretation changes a lot if perhaps only CD4s change and CD8s are unaffected. One particular important analysis that is important is the quantification of CM/EM/TEFF phenotypes, because these are all very different between CD4s and CD8s. I certainly don't need this for all the RNAseq, just the major markers analyzed by FACS.

ii) The most important places to show extra phenotype data are in the kinetics experiment in Figure 4/5 and the in vivo experiments in Fig 5. I think all the markers the authors are generally showing are what is necessary (TCF/PD1/CD39/GZMB/KLRG etc.), but please sample these at various timepoints across the timecourse analyzed and provide FACS plots of the cells in the different conditions. Stratify the cells into CD4 and CD8 so people can see any differences, use standard FACS plots from other papers the cells are being compared to. E.g TCF x Gzm, or KLRG x IL7R, show the Tcm/em plots. Show Bach2, especially in the TMP experiments and Bach2 vs. major markers to understand if there is some heterogeneity in the mixed populations. I think Bach2 x TCF and Bach2 x Gzm will give a lot of information

about what is happening.

iii) The in vivo experiments firstly need quite a lot more detail of the experimental setups, but similarly need much more detailed analysis of what happens to the cells once they are put into the cancer bearing mice. First, it is unclear to me if the TMP treatment continues in the Bach2-ON mice? If it does, does Bach2 expression stay on? If TMP is not given, I am assuming Bach2 comes down quickly. Do the phenotypes of the Bach2-On cells retain some undifferentiated phenotype even if the TMP is not given. Are the CD4/CD8 ratios different between the groups. I think ~3 timepoints after transfer with detailed FACS on the transferred cells is all that's needed. I don't think any result would be bad, but there needs to be a complete analysis of the cells to understand how the different conditions

3. I have a few technical or control issues that might be cleared up with some more explanation, or might need some experiments.

i) In the cytotoxicity assays in figure 1d/3j, the authors state that T cells are co-cultured with cancer cells at a 1:2 ratio. Is this the ratio at each time point, or was this the ratio at D0 and now the ratio is either unclear or probably different between the groups? It seems that the cells with the most expansion are the ones with the most cytotoxicity, so this assay might just be showing more T cells kill the targets better, which doesn't add much beyond the expansion data. I think people mostly want to know if on a 1 to 1 comparison, is one condition making a better killer than the other. I think just doing the killing assays at the final timepoint of the expansion protocol with the same ratio of T:Target addresses this very easily.

ii) In the in vivo experiments (5K-M) need a lot more detail of how each mouse is being treated and I think the authors need to include a group with the BACH2-OE +/- the TMP drug. As I say above, it's a little unclear to me if the TMP treatment continued after transfer of CART into the mice, but assuming it was, it could be the case that TMP is inhibiting tumor growth explaining the difference between BACH2-On and BACH2-Off. I think Bach2-OE +/- TMP will rule this out, but I could see 22/BB-S +/- TMP being a good control too. This one is up to the authors and if they didn't treat mice with TMP then no problems on the tumor growth, but as asked above, I need to see how the cells are changing in vivo.

4. Interpretation/clarification— None of these points are critical for the central claims of the paper, but I think needs some additional thought and clarification.

i) Can the authors confirm the data in Fig7 is just the pre-infusion product from the Wilson Cancer Discovery paper? If it includes the post-infusion timepoints, there needs to be a lot more clarification and discussion because Bach2 is clearly not the same in antigen +/- conditions.

ii) I know the authors know that PD1 expression is not a definitive marker of 'exhaustion', but it conveniently lines up with some of the observations they are making. PD1 in most of these in vitro conditions is a readout of TCR signal strength and there are probably some much more interesting interpretations of the PD1 data in the paper unrelated to exhaustion. Something like Fig 1C where the strongest tonic signaling receptor (28-S) has the highest PD1 expression might be telling you something about how the receptor is acting. Or that in Fig 3J/K/L there is still 50% of the cancer left in the 28-S conditions and that has the most PD1 so its not really a fair comparison. None of this is very important, but I would just suggest the authors think about their PD1 data in a little more detail than 'exhaustion'.

iii) I'm not sure if this is known, but will the e-coli derived protein tag be a target of rejection of the T cells if this is attempted in humans? rTTA is enough in mice so I would expect it might be an issue. This doesn't take away from the good work, but probably needs some discussion.

Reviewer #2

(Remarks to the Author)

This study by Chang et al. identifies BACH2 as a key transcriptional regulator activated by tonic signaling in CAR-T cells, mediating the distinct effects of CD28 and 41BB costimulatory domains on T cell phenotype, exhaustion, and function. The authors demonstrate that BACH2 promotes memory and stem-like states while preventing exhaustion. However, they also find that high BACH2 expression can constrain CAR-T cell functionality by limiting the transition to effector states, necessitating precise regulation. By linking BACH2 to a degradation domain, they achieve controlled expression, enhancing CAR-T cell persistence and efficacy. Furthermore, they establish a correlation between BACH2 activity in manufactured CAR-T products and clinical outcomes in leukemia patients, underscoring its therapeutic relevance. While the study is very interesting, additional experiments would further strengthen its conclusions.

Major comments:

- Since the study aims to establish BACH2 as a key transcriptional regulator induced by tonic signaling, a deeper epigenetic analysis is warranted. The identification of BACH2 as a transcriptional regulator in 22/BB-S tonic signaling cells is inferred from RNA sequencing data. To corroborate these findings, ATAC-seq should be performed to compare tonic versus non-tonic signaling CAR-T cells across the 2 different costimulatory domains (22/BB-S, 22/BB-L, 22/28-S, 22/28-L). Transcription factor (TF) motif enrichment analysis would help determine whether BACH2 emerges as a significantly enriched TF motif in 22/BB-S CAR-T cells. These analyses should also be conducted in both unedited and BACH2 knockout (KO) CAR-T cells to validate the results. Further, ChIP-seq is needed to identify direct BACH2 target genes and elucidate its regulatory network.
- As BACH2 has already been established as a key transcriptional and epigenetic regulator of stem-like CD8+ T cells

(PMID: 33574619), the novelty of this study lies in its connection to tonic signaling in CAR-T cells and the differential activity in CD28 and 41BB costim domains. A more detailed analysis of this association would strengthen the manuscript. Specifically, quantifying the level of tonic signaling at various time points during CAR-T cell manufacturing and correlating it with BACH2 expression at both transcriptional and proteomic levels would provide deeper insights.

• In Vivo Studies need strengthening. The study relies on a single mouse model, and the observed improvement in tumor burden is modest. Additional in vivo models should be incorporated to substantiate the findings, including solid tumor models if possible. Moreover, key questions remain unanswered: What is the expansion and persistence of CAR-T cells in vivo? What is their phenotype over time? How does BACH2 expression change in vivo, and does it correlate with persistence and central memory formation?

Specific Figure Comments:

-Figure 1: The authors assess PD-1 expression, but it would be valuable to examine additional exhaustion markers such as LAG3, TIM3, and CTLA-4.

-Figure 3g: Why does 22/BB-S not show high expression of memory-related genes (CCR7, LEF1, TCF7)?

-Figure 3n-q: Inclusion of 22/28-S as a control in these assays would help determine whether BACH2 KO specifically abrogates the advantage seen in 22/BB-S cells.

-Figure 3: Functional validation of BACH2 overexpression and KO CAR-T cells in rechallenge cytotoxicity assays would provide further insights into their long-term efficacy.

-Figure 5b-j: Including 22/BB-S as a control would help to compare BACH2 expression levels between BACH2DD OFF and 22/BB-S, as well as their corresponding phenotypic and functional characteristics.

Reviewer #3

(Remarks to the Author)

In this manuscript, the authors evaluate differences between CAR-T cells expressing different CD28 and 4-1BB internal domains. The authors had previously compared a relatively successful anti-CD22 CAR-T cells expressing a 4-1BB chimera constructed with a short linker (22/BB-S), showing increased tonic signaling associated with receptor clustering and activation of PI3K and MAPK pathways than those expressing a chimeric receptor with a long linker that were less successful in vivo (ref 6). They now compare two 4-1BB-based CARs with either a short (22/BB-S) or long (22/BB-L) linker to those with similar linkers using a CD28 chimera, finding that the 22/BB-S provided the best functionality and least signs of exhaustion. The CD28 chimeras showed the opposite trend (22/28-L > 22/28S). They show that differences among these receptors are associated with a BACH2 transcriptional signature (among others), with the more effective 4-1BB short chimera (22/BB-S) showing the greatest expression of BACH2, a transcription factor that has been shown to promote progenitor stem-like CD8 cells during exhaustion. They evaluate the effects of overexpression of Bach2 in the 22/28-S line, showing that this increases TCF1+ cells and decreased PD-1+ cells, yet does not increase expansion or generation of effectors, suggesting it locks T cells in a progenitor state. Through the generation of a degradable form of Bach2 which they could express at low levels in the absence of Trimethoprim (TMP), the authors show improved cell expansion and function in 22/28S CAR T cells that they engineer to express low levels of Bach2, with increased anti-tumor activity in a xenograft model. Finally, evaluation of data from a number of clinical trials showed a correlation between Bach2 levels and patient survival. The paper contributes to a growing recognition that tunable levels of certain transcription factors, such as FoxO1 and now BACH2 may improve stemness, while permitting effector function. However, it is not clear how widely applicable their findings are, given that expression of low levels of BACH2 appear to work primarily in one setting.

Specific comments:

The authors need to be clear when talking about tonic signaling from 4-1BB versus CD28 versus tonic signaling in general. This difference is unclear in multiple places in the paper (e.g. paragraph 2 of intro). Different receptors may have different types of tonic signaling and the authors need to be clear in their descriptions. A description of their previous findings on the short and long linker version of the anti-CD22-4-1BB CAR would be useful as well as signaling observed with the CD28 chimeras.

In Fig 5m, what happens to the function if there is a titration of TMP?

In the xenograft model, based on the data in Ext Data 4B, one would expect BACH2 levels to drop after a day after transfer, yet, it seems as if a subsequent reduction of BACH2 levels does not permit effector cell differentiation. Is it possible to check the BACH2 levels in the CAR-T cells several days after transfer in the xenograft model?

Based on the GD-2 model, the amount of Bach2 protein required to promote stemness and influence Tem appears to depend on the CAR-T construct (although the functionality is not tested). This makes the utility of the technique more questionable, although this is an important observation. Did the authors test the function of the GD-2 cells expressing BACH2? What happens if BACH2-OFF is expressed in the 22/BB-S (and 22/BB-L) expressing cells? Does the small amount of BACH2 improve their subsequent function? Even in the context of the 22/28-S construct, did the authors try a titration of BACH2?

In Fig 7d, BACH2 scores in CD4 Tcm correlates best with improved survival and no relapse—do the authors have further insight into this from other information on these cells. Are the CD4 cells cytolytic or do the effects appear secondary to/correlate with cytokine production that might promote CD8 cell function?

Minor points:

I think Ext Data 1c is miscited as 1e.

This is an interesting paper on use of a BACH2 degenon construct to regulate BACH2 levels and improve CAR-T function. I realize that this is a lot of work with human CAR-T cells, but while they show low BACH2 improves function of one CAR-T,

and the deGron approach is strong, it is not clear how generalizable their findings may be (for function). Perhaps the strongest support of the paper are the correlations of survival/relapse-free responses with the in vivo Bach2 signature.

Decision Letter:

21st Feb 2025

Dear Dr Singh,

As you know, your Article, "BACH2 regulates T cell lineage states to overcome dysfunction driven by tonic CAR signaling" has now been seen by 3 referees and from their comments copied below they find your work of considerable potential interest, but have raised quite substantial concerns that must be addressed.

We have now also looked over your Author Response to these comments and we are pleased to say that the revision plan looks very good to us, so please do go ahead and begin the revision. We can schedule a VC if you need.

If you choose to revise your manuscript taking into account all reviewer and editor comments, please highlight all changes in the manuscript text file in Microsoft Word format.

* If you have not done so already please begin to revise your manuscript so that it conforms to our Article format instructions at <http://www.nature.com/ni/authors/index.html>. Refer also to any guidelines provided in this letter.

The Reporting Summary can be found here:

Extended Data figures and tables are online-only (appearing in the online PDF and full-text HTML version of the paper), peer-reviewed display items that provide essential background to the Article but are not included in the printed version of the paper due to space constraints or being of interest only to a few specialists. A maximum of ten Extended Data display items (figures and tables) is typically permitted. When re-submitting your manuscript, please ensure that any supplementary figures and tables that are more critical to the manuscript's conclusions are converted to Extended data to increase these data's visibility.

Link Redacted

If you wish to submit a suitably revised manuscript we would hope to receive it within 6 months. If you cannot send it within this time, please let us know. We will be happy to consider your revision so long as nothing similar has been accepted for publication at Nature Immunology or published elsewhere.

Nature Immunology is committed to improving transparency in authorship. As part of our efforts in this direction, we are now requesting that all authors identified as 'corresponding author' on published papers create and link their Open Researcher and Contributor Identifier (ORCID) with their account on the Manuscript Tracking System (MTS), prior to acceptance. ORCID helps the scientific community achieve unambiguous attribution of all scholarly contributions. You can create and link your ORCID from the home page of the MTS by clicking on 'Modify my Springer Nature account'. For more information please visit www.springernature.com/orcid.

Thank you for the opportunity to review your work.

Sincerely,

Nick Bernard, PhD
Senior Editor
Nature Immunology

Reviewers' Comments:

Reviewer #1 (Remarks to the Author):

In this work, the authors study the effect of over-expression of BACH2 on the functionality of T cells being expanded for CAR therapy *in vitro*, and how this molecule contributes to the anti-tumor killing ability of the CART cells. The authors find that if they force expression of BACH2, they can increase the expansion potential of CD28 tailed CARs, which usually have the problem of crashing after a short time *in vitro*. The challenge of just over-expressing Bach2 is that this molecule inhibits effector differentiation, so the cells are actually worse *in vivo*. However, the big highlight of the work is the authors developed a very novel method to allow timed deletion of the Bach2 molecule using an *ecoli* derived degradation domain that allowed them to turn Bach2 on and off with an antibiotic. Using this approach they take all the benefits of expressing Bach2 during expansion, but then using their novel system, turn it off by withdrawing the antibiotic. Overall, this is a very clever system to allow timed expression of BACH2 to maximize its inhibitory role during expansion, but then remove the effect when the cells need to take on an effector state. The idea is novel and builds on strong previous fundamental findings related to the role of BACH2 in T cell function and certainly contributes to new ideas on how to use CAR T cells in cancer. My biggest issue is that there are quite a lot of places in the current version of the manuscript that need additional analysis of the cells to provide better understanding of how BACH2 is altering the phenotype and function *in vitro* and *in vivo* over time and some more detail around experimental setups and interpretation.

1. I expect the authors have a lot of these data, and I don't fault them at all, but throughout the paper there needs to be inclusion of representative FACS plots of all markers examined. Other labs will need to repeat this work and will need to see what the authors are presenting so they can see if the observations are reproducible. All FACS plots need to show outlier dots if they are contours, axis ticks with the scale, and the fluorescent color/metal of the marker used. I have no doubts about the authors claims, but readers need to be able to compare to what they find in their experiments or what is seen in other pieces of work.

2. The major experimental additions that are needed are all related to providing more detailed analysis of the T cells in the different conditions tested. The authors are generally making a lot of claims about T cell exhaustion and other states. They have some excellent functional data like tumor killing or the ability of the cells to expand, but there is very little detail of how the cells respond over time or *in vivo*.

i) Throughout the paper, the authors need to specify if they are looking at CD4 or CD8 T cells when they are showing phenotypic changes. There are good reasons to think that CD4 and CD8 T cells might not respond in the same way to BACH2 overexpression and the interpretation changes a lot if perhaps only CD4s change and CD8s are unaffected. One particular important analysis that is important is the quantification of CM/EM/TEFF phenotypes, because these are all very different between CD4s and CD8s. I certainly don't need this for all the RNAseq, just the major markers analyzed by FACS.

ii) The most important places to show extra phenotype data are in the kinetics experiment in Figure 4/5 and the *in vivo* experiments in Fig 5. I think all the markers the authors are generally showing are what is necessary (TCF/PD1/CD39/GZMB/KLRG etc.), but please sample these at various timepoints across the timecourse analyzed and provide FACS plots of the cells in the different conditions. Stratify the cells into CD4 and CD8 so people can see any differences, use standard FACS plots from other papers the cells are being compared to. E.g TCF x Gzm, or KLRG x IL7R, show the Tcm/em plots. Show Bach2, especially in the TMP experiments and Bach2 vs. major markers to understand if there is some heterogeneity in the mixed populations. I think Bach2 x TCF and Bach2 x Gzm will give a lot of information about what is happening.

iii) The *in vivo* experiments firstly need quite a lot more detail of the experimental setups, but similarly need much more detailed analysis of what happens to the cells once they are put into the cancer bearing mice. First, it is unclear to me if the TMP treatment continues in the Bach2-ON mice? If it does, does Bach2 expression stay on? If TMP is not given, I am assuming Bach2 comes down quickly. Do the phenotypes of the Bach2-On cells retain some undifferentiated phenotype

even if the TMP is not given. Are the CD4/CD8 ratios different between the groups. I think ~3 timepoints after transfer with detailed FACS on the transferred cells is all that's needed. I don't think any result would be bad, but there needs to be a complete analysis of the cells to understand how the different conditions

3. I have a few technical or control issues that might be cleared up with some more explanation, or might need some experiments.

i) In the cytotoxicity assays in figure 1d/3j, the authors state that T cells are co-cultured with cancer cells at a 1:2 ratio. Is this the ratio at each time point, or was this the ratio at D0 and now the ratio is either unclear or probably different between the groups? It seems that the cells with the most expansion are the ones with the most cytotoxicity, so this assay might just be showing more T cells kill the targets better, which doesn't add much beyond the expansion data. I think people mostly want to know if on a 1 to 1 comparison, is one condition making a better killer than the other. I think just doing the killing assays at the final timepoint of the expansion protocol with the same ratio of T:Target addresses this very easily.

ii) In the in vivo experiments (5K-M) need a lot more detail of how each mouse is being treated and I think the authors need to include a group with the BACH2-OE +/- the TMP drug. As I say above, it's a little unclear to me if the TMP treatment continued after transfer of CART into the mice, but assuming it was, it could be the case that TMP is inhibiting tumor growth explaining the difference between BACH2-On and BACH2-Off. I think Bach2-OE +/- TMP will rule this out, but I could see 22/BB-S +/- TMP being a good control too. This one is up to the authors and if they didn't treat mice with TMP then no problems on the tumor growth, but as asked above, I need to see how the cells are changing in vivo.

4. Interpretation/clarification— None of these points are critical for the central claims of the paper, but I think needs some additional thought and clarification.

i) Can the authors confirm the data in Fig7 is just the pre-infusion product from the Wilson Cancer Discovery paper? If it includes the post-infusion timepoints, there needs to be a lot more clarification and discussion because Bach2 is clearly not the same in antigen +/- conditions.

ii) I know the authors know that PD1 expression is not a definitive marker of 'exhaustion', but it conveniently lines up with some of the observations they are making. PD1 in most of these in vitro conditions is a readout of TCR signal strength and there are probably some much more interesting interpretations of the PD1 data in the paper unrelated to exhaustion. Something like Fig 1C where the strongest tonic signaling receptor (28-S) has the highest PD1 expression might be telling you something about how the receptor is acting. Or that in Fig 3J/K/L there is still 50% of the cancer left in the 28-S conditions and that has the most PD1 so its not really a fair comparison. None of this is very important, but I would just suggest the authors think about their PD1 data in a little more detail than 'exhaustion'.

iii) I'm not sure if this is known, but will the e-coli derived protein tag be a target of rejection of the T cells if this is attempted in humans? rTTa is enough in mice so I would expect it might be an issue. This doesn't take away from the good work, but probably needs some discussion.

Reviewer #2 (Remarks to the Author):

This study by Chang et al. identifies BACH2 as a key transcriptional regulator activated by tonic signaling in CAR-T cells, mediating the distinct effects of CD28 and 41BB costimulatory domains on T cell phenotype, exhaustion, and function. The authors demonstrate that BACH2 promotes memory and stem-like states while preventing exhaustion. However, they also find that high BACH2 expression can constrain CAR-T cell functionality by limiting the transition to effector states, necessitating precise regulation. By linking BACH2 to a degradation domain, they achieve controlled expression, enhancing CAR-T cell persistence and efficacy. Furthermore, they establish a correlation between BACH2 activity in manufactured CAR-T products and clinical outcomes in leukemia patients, underscoring its therapeutic relevance. While the study is very interesting, additional experiments would further strengthen its conclusions.

Major comments:

- Since the study aims to establish BACH2 as a key transcriptional regulator induced by tonic signaling, a deeper epigenetic analysis is warranted. The identification of BACH2 as a transcriptional regulator in 22/BB-S tonic signaling cells is inferred from RNA sequencing data. To corroborate these findings, ATAC-seq should be performed to compare tonic versus non-tonic signaling CAR-T cells across the 2 different costimulatory domains (22/BB-S, 22/BB-L, 22/28-S, 22/28-L). Transcription factor (TF) motif enrichment analysis would help determine whether BACH2 emerges as a significantly enriched TF motif in 22/BB-S CAR-T cells. These analyses should also be conducted in both unedited and BACH2 knockout (KO) CAR-T cells to validate the results. Further, ChIP-seq is needed to identify direct BACH2 target genes and elucidate its regulatory network.
- As BACH2 has already been established as a key transcriptional and epigenetic regulator of stem-like CD8+ T cells (PMID: 33574619), the novelty of this study lies in its connection to tonic signaling in CAR-T cells and the differential activity in CD28 and 41BB costim domains. A more detailed analysis of this association would strengthen the manuscript. Specifically, quantifying the level of tonic signaling at various time points during CAR-T cell manufacturing and correlating it with BACH2 expression at both transcriptional and proteomic levels would provide deeper insights.
- In Vivo Studies need strengthening. The study relies on a single mouse model, and the observed improvement in tumor burden is modest. Additional in vivo models should be incorporated to substantiate the findings, including solid tumor

models if possible. Moreover, key questions remain unanswered: What is the expansion and persistence of CAR-T cells in vivo? What is their phenotype over time? How does BACH2 expression change in vivo, and does it correlate with persistence and central memory formation?

Specific Figure Comments:

- Figure 1: The authors assess PD-1 expression, but it would be valuable to examine additional exhaustion markers such as LAG3, TIM3, and CTLA-4.
- Figure 3g: Why does 22/BB-S not show high expression of memory-related genes (CCR7, LEF1, TCF7)?
- Figure 3n-q: Inclusion of 22/28-S as a control in these assays would help determine whether BACH2 KO specifically abrogates the advantage seen in 22/BB-S cells.
- Figure 3: Functional validation of BACH2 overexpression and KO CAR-T cells in rechallenge cytotoxicity assays would provide further insights into their long-term efficacy.
- Figure 5b-j: Including 22/BB-S as a control would help to compare BACH2 expression levels between BACH2DD OFF and 22/BB-S, as well as their corresponding phenotypic and functional characteristics.

Reviewer #3 (Remarks to the Author):

In this manuscript, the authors evaluate differences between CAR-T cells expressing different CD28 and 4-1BB internal domains. The authors had previously compared a relatively successful anti-CD22 CAR-T cells expressing a 4-1BB chimera constructed with a short linker (22/BB-S), showing increased tonic signaling associated with receptor clustering and activation of PI3K and MAPK pathways than those expressing a chimeric receptor with a long linker that were less successful in vivo (ref 6). They now compare two 4-1BB-based CARs with either a short (22/BB-S) or long (22/BB-L) linker to those with similar linkers using a CD28 chimera, finding that the 22/BB-S provided the best functionality and least signs of exhaustion. The CD28 chimeras showed the opposite trend (22/28-L > 22/28-S). They show that differences among these receptors are associated with a BACH2 transcriptional signature (among others), with the more effective 4-1BB short chimera (22/BB-S) showing the greatest expression of BACH2, a transcription factor that has been shown to promote progenitor stem-like CD8 cells during exhaustion. They evaluate the effects of overexpression of Bach2 in the 22/28-S line, showing that this increases TCF1+ cells and decreased PD-1+ cells, yet does not increase expansion or generation of effectors, suggesting it locks T cells in a progenitor state. Through the generation of a degradable form of Bach2 which they could express at low levels in the absence of Trimethoprim (TMP), the authors show improved cell expansion and function in 22/28S CAR T cells that they engineer to express low levels of Bach2, with increased anti-tumor activity in a xenograft model. Finally, evaluation of data from a number of clinical trials showed a correlation between Bach2 levels and patient survival. The paper contributes to a growing recognition that tunable levels of certain transcription factors, such as FoxO1 and now BACH2 may improve stemness, while permitting effector function. However, it is not clear how widely applicable their findings are, given that expression of low levels of BACH2 appear to work primarily in one setting.

Specific comments:

The authors need to be clear when talking about tonic signaling from 4-1BB versus CD28 versus tonic signaling in general. This difference is unclear in multiple places in the paper (e.g. paragraph 2 of intro). Different receptors may have different types of tonic signaling and the authors need to be clear in their descriptions. A description of their previous findings on the short and long linker version of the anti-CD22-4-1BB CAR would be useful as well as signaling observed with the CD28 chimeras.

In Fig 5m, what happens to the function if there is a titration of TMP?

In the xenograft model, based on the data in Ext Data 4B, one would expect BACH2 levels to drop after a day after transfer, yet, it seems as if a subsequent reduction of BACH2 levels does not permit effector cell differentiation. Is it possible to check the BACH2 levels in the CAR-T cells several days after transfer in the xenograft model?

Based on the GD-2 model, the amount of Bach2 protein required to promote stemness and influence Tem appears to depend on the CAR-T construct (although the functionality is not tested). This makes the utility of the technique more questionable, although this is an important observation. Did the authors test the function of the GD-2 cells expressing BACH2? What happens if BACH2-OFF is expressed in the 22/BB-S (and 22/BB-L) expressing cells? Does the small amount of BACH2 improve their subsequent function? Even in the context of the 22/28-S construct, did the authors try a titration of BACH2?

In Fig 7d, BACH2 scores in CD4 Tcm correlates best with improved survival and no relapse—do the authors have further insight into this from other information on these cells. Are the CD4 cells cytolytic or do the effects appear secondary to/correlate with cytokine production that might promote CD8 cell function?

Minor points:

I think Ext Data 1c is miscited as 1e.

This is an interesting paper on use of a BACH2 degron construct to regulate BACH2 levels and improve CAR-T function. I realize that this is a lot of work with human CAR-T cells, but while they show low BACH2 improves function of one CAR-T, and the degron approach is strong, it is not clear how generalizable their findings may be (for function). Perhaps the strongest support of the paper are the correlations of survival/relapse-free responses with the in vivo Bach2 signature.

Version 1:

Reviewer comments:

Reviewer #1

(Remarks to the Author)

In the previous version of the paper I had asked for more detail of the in vivo phenotype of the Bach2 CART and most importantly how these cells changed over time. In addition, I had asked for much more detail about the phenotype of the cells and if the BACH2 program was stable after cessation of treatment. The authors have addressed all my points with new data and clearly demonstrated the details I needed to see about these cells.

Reviewer #2

(Remarks to the Author)

The authors thoroughly addressed the reviewers' comments and provided new data that significantly strengthened the manuscript. In instances where additional experiments were not feasible due to technical limitations, they clearly demonstrated the efforts undertaken and offered a sound rationale for why the resulting data could not be incorporated. Regarding figure-specific comment 4, the authors included the rechallenge data with BACH2 knockout in their response (reviewer Figure 1). This dataset supports the authors' hypothesis, and I would prefer to see it included in the manuscript itself.

Reviewer #3

(Remarks to the Author)

The authors have addressed most of my questions. The paper has many important and strong points including the titratable or ON/off BACH-2 construct, the demonstration that the amount of BACH-2 required depends on the CAR-T and the correlation of BACH-2 levels with successful CAR-T therapy. These are important contributions. However, I find the paper difficult to read, primarily because of the emphasis on tonic signaling, particularly in the beginning of the paper. I would like to make a minor suggestion that would help me (and I think other readers).

A suggestion: the authors start out trying to evaluate the effects of altered tonic signaling by altering svFc linker length "with either long (20 residue, no tonic signaling) or short (5 residue, tonic signaling) linkers and containing either CD28 or 41BB costimulatory domains". The 22/BB-L construct was less effective under a number of parameters than the short linker (22/BB-S) construct. But the short version of the 22/28 construct was less effective than the long version, so it is hard to draw conclusions about short vs long linkers (and tonic signaling) in this context. It would be so much easier if, at this point, the authors just stated that the 22/BB-S construct was the best and that they decided to look to see what distinguished this construct from the others at the gene expression level (and not talk about tonic signaling anymore, which was confusing and is distracting from the main point of the paper). For example, at the start of the second section, could they change to something like:

"22/BB-S signaling induces BACH2

To determine the molecular etiology of these differences in CAR-T function, we interrogated differences in gene expression between 22/BB-S and the other constructs using RNA sequencing, focusing on differences with 22/28-S."

I realize the authors came to this work from the perspective of tonic signaling but I think it just confuses their very strong message about BACH2. I think this would simplify the message for the readers and lead more clearly and strongly into the heart of the paper. Similarly, they might want to slightly modify their description of the HA-GD2 CAR. These are really only minor changes, but would make the paper more cohesive.

Other minor points:

When the authors talk about their chronic stimulation cultures, could the authors clarify how these are different from the CAR-T manufacturing conditions? Do the chronic stimulation cultures start after the generation of the CAR-T cultures or is it a difference in how often stimulating cells are added. A sentence will suffice.

From their ATAC-Seq data, it does look like BACH2-OE decreases c-Jun binding in the 22/28-S construct. However, the 22/BB-S construct has more c-Jun binding. The authors may want to comment on this, as that it may not be the only difference.

Besides these points, I think the paper has many interesting and important points.

Decision Letter:

Our ref: NI-A39520A

11th Nov 2025

Dear Dr. Singh,

Thank you for submitting your revised manuscript "BACH2 regulates T cell lineage states to overcome dysfunction driven by tonic CAR signaling" (NI-A39520A). It has now been seen by the original referees and their comments are below. The reviewers find that the paper has improved in revision, and therefore we'll be happy in principle to publish it in Nature Immunology, pending minor revisions to satisfy the referees' final requests and to comply with our editorial and formatting guidelines.

We will now perform detailed checks on your paper and will send you a checklist detailing our editorial and formatting requirements in about a week. Please do not upload the final materials and make any revisions until you receive this additional information from us.

If you had not uploaded a Word file for the current version of the manuscript, we will need one before beginning the editing process; please email that to immunology@us.nature.com at your earliest convenience.

Thank you again for your interest in Nature Immunology Please do not hesitate to contact me if you have any questions.

Sincerely,

Nick Bernard, PhD
Senior Editor
Nature Immunology

Reviewer #1 (Remarks to the Author):

In the previous version of the paper I had asked for more detail of the in vivo phenotype of the Bach2 CART and most importantly how these cells changed over time. In addition, I had asked for much more detail about the phenotype of the cells and if the BACH2 program was stable after cessation of treatment. The authors have addressed all my points with new data and clearly demonstrated the details I needed to see about these cells.

Reviewer #2 (Remarks to the Author):

The authors thoroughly addressed the reviewers' comments and provided new data that significantly strengthened the manuscript. In instances where additional experiments were not feasible due to technical limitations, they clearly demonstrated the efforts undertaken and offered a sound rationale for why the resulting data could not be incorporated. Regarding figure-specific comment 4, the authors included the rechallenge data with BACH2 knockout in their response (reviewer Figure 1). This dataset supports the authors' hypothesis, and I would prefer to see it included in the manuscript itself.

Reviewer #3 (Remarks to the Author):

The authors have addressed most of my questions. The paper has many important and strong points including the titratable or ON/off BACH-2 construct, the demonstration that the amount of BACH-2 required depends on the CAR-T and the correlation of BACH-2 levels with successful CAR-T therapy. These are important contributions. However, I find the paper difficult to read, primarily because of the emphasis on tonic signaling, particularly in the beginning of the paper. I would like to make a minor suggestion that would help me (and I think other readers).

A suggestion: the authors start out trying to evaluate the effects of altered tonic signaling by altering svFc linker length "with either long (20 residue, no tonic signaling) or short (5 residue, tonic signaling) linkers and containing either CD28 or 41BB costimulatory domains". The 22/BB-L construct was less effective under a number of parameters than the short linker (22/BB-S) construct. But the short version of the 22/28 construct was less effective than the long version, so it is hard to draw conclusions about short vs long linkers (and tonic signaling) in this context. It would be so much easier if, at this point, the authors just stated that the 22/BB-S construct was the best and that they decided to look to see what distinguished this construct from the others at the gene expression level (and not talk about tonic signaling anymore, which was confusing and is distracting from the main point of the paper). For example, at the start of the second section, could they change to something like:

"22/BB-S signaling induces BACH2

To determine the molecular etiology of these differences in CAR-T function, we interrogated differences in gene expression between 22/BB-S and the other constructs using RNA sequencing, focusing on differences with 22/28-S."

I realize the authors came to this work from the perspective of tonic signaling but I think it just confuses their very strong message about BACH2. I think this would simplify the message for the readers and lead more clearly and strongly into the heart of the paper. Similarly, they might want to slightly modify their description of the HA-GD2 CAR. These are really only minor changes, but would make the paper more cohesive.

Other minor points:

When the authors talk about their chronic stimulation cultures, could the authors clarify how these are different from the CAR-T manufacturing conditions? Do the chronic stimulation cultures start after the generation of the CAR-T cultures or is it a difference in how often stimulating cells are added. A sentence will suffice.

From their ATAC-Seq data, it does look like BACH2-OE decreases c-Jun binding in the 22/28-S construct. However, the 22/BB-S construct has more c-Jun binding. The authors may want to comment on this, as that it may not be the only difference.

Besides these points, I think the paper has many interesting and important points.

Responses to Review

We greatly thank the reviewers for their largely positive, thorough and thoughtful comments about our paper. The addition of these new data, narrative clarifications and experimental details has greatly strengthened our conclusions and given us even more confidence about the central role of BACH2 on CAR T cell function. We are very grateful for these highly constructive suggestions which have led to a much more cohesive manuscript. Please note that all changes to manuscript text are in blue.

Reviewer #1

General

In this work, the authors study the effect of over-expression of BACH2 on the functionality of T cells being expanded for CAR therapy in vitro, and how this molecule contributes to the anti-tumor killing ability of the CART cells. The authors find that if they force expression of BACH2, they can increase the expansion potential of CD28 tailed CARs, which usually have the problem of crashing after a short time in vitro. The challenge of just over-expressing Bach2 is that this molecule inhibits effector differentiation, so the cells are actually worse in vivo. However, the big highlight of the work is the authors developed a very novel method to allow timed deletion of the Bach2 molecule using an ecoli derived degradation domain that allowed them to turn Bach2 on and off with an antibiotic. Using this approach they take all the benefits of expressing Bach2 during expansion, but then using their novel system, turn it off by withdrawing the antibiotic.

Overall, this is a very clever system to allow timed expression of BACH2 to maximize its inhibitory role during expansion, but then remove the effect when the cells need to take on an effector state. The idea is novel and builds on strong previous fundamental findings related to the role of BACH2 in T cell function and certainly contributes to new ideas on how to use CAR T cells in cancer. My biggest issue is that there are quite a lot of places in the current version of the manuscript that need additional analysis of the cells to provide better understanding of how BACH2 is altering the phenotype and function in vitro and in vivo over time and some more detail around experimental setups and interpretation.

Comment 1

I expect the authors have a lot of these data, and I don't fault them at all, but throughout the paper there needs to be inclusion of representative FACS plots of all markers examined. Other labs will need to repeat this work and will need to see what the authors are presenting so they can see if the observations are reproducible. All FACS plots need to show outlier dots if they are contours, axis ticks with the scale, and the fluorescent color/metal of the marker used. I have no doubts about the authors claims, but readers need to be able to compare to what they find in their experiments or what is seen in other pieces of work.

Response

Our apologies for not including these important metrics in the original manuscript. We have overhauled our data presentation to include representative flow plots throughout this revised manuscript, including our flow cytometry gating strategy (**Extended Data Fig. 10**) and representative plots for all major markers (**Figures 3m, 4g, Extended Data Figs. 3a, 4 and 6**). These new plots (and modifications to previous plots) now include original axes, outliers, and details of marker used for detection. These are accompanied by **Supplementary Tables 1-2** which have manufacturer and clone details for each antibody.

Comment 2.1

The major experimental additions that are needed are all related to providing more detailed analysis of the T cells in the different conditions tested. The authors are generally making a lot of claims about T cell exhaustion and other states. They have some excellent functional data like tumor killing or the ability of the cells to expand, but

there is very little detail of how the cells respond over time or *in vivo*. Throughout the paper, the authors need to specify if they are looking at CD4 or CD8 T cells when they are showing phenotypic changes. There are good reasons to think that CD4 and CD8 T cells might not respond in the same way to BACH2 overexpression and the interpretation changes a lot if perhaps only CD4s change and CD8s are unaffected. One particular important analysis that is important is the quantification of CM/EM/TEFF phenotypes, because these are all very different between CD4s and CD8s. I certainly don't need this for all the RNAseq, just the major markers analyzed by FACS.

Response

We too were interested in any differential activity of CD4 or CD8 cells, and for this reason included these markers in all of our spectral flow cytometry studies. Our omission of them in the original manuscript was because of the minimal differences observed between subsets. We certainly appreciate that natural CD4 and CD8s follow distinct pathways of differentiation in response to stimulation; however, our experience in CAR engineering of human cells is that, in large part, the process of activation through a CAR does not yield the same differences in memory differentiation. As noted below, the glaring exception is expression of cytotoxic proteins, which we nearly always find to be higher in CD8 CAR T cells.

Specifically to the points raised in this comment, we have now broken experiments out by CD4, CD8 and bulk T cell populations, performed the requested analyses, included representative plots and added additional time points. These new data include:

- BACH2 expression at the conclusion of manufacturing and end of chronic stimulation
- Memory phenotype at the end of manufacturing and at all evaluated time points of chronic stimulation
- Additional timepoints of analysis for CAR T cell phenotypes during chronic stimulation
- Expression of CD127/KLRG1 and TCF1/GZMB throughout chronic stimulation
- Memory phenotype at the end of manufacturing cells with BACH2-degrons
- Ratio of CD4/CD8 cells *in vitro* and *in vivo*

These data are presented throughout the manuscript, but the salient observations from these studies were:

- BACH2 expression, both endogenous and transgenic, is equivalent in CD4 and CD8s throughout the experiments conducted (**Extended Data Fig. 2a** and **Extended Data Fig. 3c**).
- CD4+ cells with higher BACH2 expression (22/BB-S and 22/28-S+BACH2^{OE}) demonstrate a slight enrichment of T_{CM} at the expense of T_{EM}. We do not think this to be of significant functional relevance, however, as the maximum difference observed between CD4 and CD8 cells is ~8% (**Extended Data Fig. 3b**). Thus, we conclude that memory differentiation is largely equivalent for CD4 and CD8 CAR T cells.
- While expression of memory markers (CD62L, CD45RO, CD127 and TCF1) does not differ substantively between CD4 and CD8 cells, we do observe, as predicted, that CD8s express much higher levels of GZMB and KLRG1 (**Extended Data Fig. 4**).
- BACH2 over-expression results in a dose-dependent enrichment of CD4 cells at the conclusion of manufacturing, but which is lost both *in vitro* and *in vivo* (**Extended Data Fig. 5e** and **7f**). We find this specific observation particularly interesting in light of our data from clinical samples showing a correlation between CD4 central memory cells, BACH2 regulon expression and overall survival (**Figure 7**). As in our *in vitro* experiments, CD4 T_{CM} cells from patients are not cytotoxic but associate with enhanced CAR T cell efficacy. Collectively, we conclude that BACH2 promotes T_{CM} and, to a lesser extent and independently, CD4 states and that both of these likely restrain long-term functionality. We expand on this observation in the **Discussion**.

Comment 2.2

Show Bach2, especially in the TMP experiments and Bach2 vs. major markers to understand if there is some heterogeneity in the mixed populations. I think Bach2 x TCF and Bach2 x Gzm will give a lot of information about what is happening.

Response

We too have been very interested in directly evaluating BACH2 and other intracellular markers simultaneously. Unfortunately, we have been unsuccessful in staining BACH2 in the same cells as other intracellular markers. Our BACH2 staining protocol requires very high quantities of BACH2 antibody, we believe due to the low abundance of BACH2 in human T cells. The only commercially available antibody is conjugated to PE (BioLegend #695604) and we have encountered very high background fluorescence, obfuscating interpretation of other analytes, despite our efforts to titrate our antibody staining cocktail. To overcome this and indirectly answer the Reviewer's question, we have correlated BACH2 intensity with the expression of TCF1, GZMB and Eomes in cells stained in parallel. We find a direct correlation between BACH2 quantity and quantity of TCF1 and Eomes and, intriguingly, that BACH2 quantity only impacts expression of GZMB at the highest levels of transgenic BACH2 overexpression. These findings are consistent with our observation that 22/28-S+BACH2^{DD} OFF enables robust cytotoxic function and are presented in **Extended Data Figs. 5g-i**.

Comment 2.3

The in vivo experiments firstly need quite a lot more detail of the experimental setups, but similarly need much more detailed analysis of what happens to the cells once they are put into the cancer bearing mice. First, it is unclear to me if the TMP treatment continues in the Bach2-ON mice? If it does, does Bach2 expression stay on?

Response

We apologize that the *in vivo* experimental details were not sufficient in the original submission; we have added text in the **Results on page 7** and in the **Methods on pages 14-15** to clarify the experimental setup and approach. We also regret that our protocol related to TMP was not clear. **Figure 5j** serves as a schematic to demonstrate that TMP was not given after CAR T cells infused. The "ON" and "OFF" conditions refer to the presence of TMP in manufacturing cultures only, as in our *in vitro* studies. We specifically clarified this on **page 7** to highlight this important point.

Comment 2.4 (continued)

If TMP is not given, I am assuming Bach2 comes down quickly. Do the phenotypes of the Bach2-On cells retain some undifferentiated phenotype even if the TMP is not given. Are the CD4/CD8 ratios different between the groups. I think ~3 timepoints after transfer with detailed FACS on the transferred cells is all that's needed. I don't think any result would be bad, but there needs to be a complete analysis of the cells to understand how the different conditions

Response

This is an intriguing question, which we had only made assumptions about based on our *in vitro* data. We had originally not bled mice to analyze CAR T cell populations after transfer. In our experience, cells sufficient for detailed analysis (>50 cells/uL) can only be collected in one time window (5-9 days) from the peripheral blood. To address the Reviewer's question, we re-established the *in vivo* experiment and sacrificed mice on days 7 and 13 (n=5 mice per group at each time point) to allow us to evaluate a higher quantity of T cells. As the Reviewer predicts, the absence of TMP *in vivo* results in reduction of BACH2 in the 22/28-S+BACH2^{DD} ON cells to the level seen in the OFF cells (**Figure 5m**). Indeed, as we observed *in vitro*, we do see that BACH2^{DD} ON cells retain higher levels of T_{CM} cells and lower levels of T_{EM} cells than BACH2^{DD} OFF and BACH2 WT cells (**Figures 5o-p**). As mentioned above, we did see a skewing towards CD4 cells at the conclusion of manufacturing for all cells that exogenous BACH2, but this trend does not hold over time (**Extended Data Fig. 7f**).

Comment 3.1

I have a few technical or control issues that might be cleared up with some more explanation, or might need some experiments. In the cytotoxicity assays in figure 1d/3j, the authors state that T cells are co-cultured with cancer cells at a 1:2 ratio. Is this the ratio at each time point, or was this the ratio at D0 and now the ratio is either unclear or probably different between the groups? It seems that the cells with the most expansion are the ones with the most cytotoxicity, so this assay might just be showing more T cells kill the targets better, which doesn't add much beyond the expansion data. I think people mostly want to know if on a 1 to 1 comparison, is one condition making a better killer than the other. I think just doing the killing assays at the final timepoint of the expansion protocol with the same ratio of T:Target addresses this very easily.

Response

The Reviewer correctly notes that, for these acute cytotoxicity assays, the E:T ratio is established at day 0 and then cancer cell survival is monitored over time without subsequent addition of cancer cells. These assays are established at the conclusion of CAR T cell manufacturing. Thus, the E:T ratio is dynamic over time and, again as the Reviewer states, reflects a combination of CAR T cell expansion and cytotoxic function. For this reason, cancer cell survival (measured by the absolute number of cancer cells in the co-culture well compared to a control well which does not have any CAR T cells) is the standard measure of function in acute cytotoxicity assays, and not the dynamic E:T ratio. The absolute number of CAR T cells and cancer cells at day 0 is the exact the same (in these experiments 0.5×10^6 cancer cells and 0.25×10^6 CAR T cells) across conditions. The design of this assay is very standard in the field and is, we believe, the exact experiment the reviewer describes (*"I think people mostly want to know if on a 1 to 1 comparison, is one condition making a better killer than the other. I think just doing the killing assays at the final timepoint of the expansion protocol with the same ratio of T:Target addresses this very easily"*). We regret this wasn't clear – we have tried to clarify the experimental design in the **Methods on page 14**.

Comment 3.2

In the in vivo experiments (5K-M) need a lot more detail of how each mouse is being treated and I think the authors need to include a group with the BACH2-OE +/- the TMP drug. As I say above, it's a little unclear to me if the TMP treatment continued after transfer of CART into the mice, but assuming it was, it could be the case that TMP is inhibiting tumor growth explaining the difference between BACH2-On and BACH2-Off. I think Bach2-OE +/- TMP will rule this out, but I could see 22/BB-S +/- TMP being a good control too. This one is up to the authors and if they didn't treat mice with TMP then no problems on the tumor growth, but as asked above, I need to see how the cells are changing in vivo.

Response

Our apologies again – TMP was not given to mice in any condition. We hope this clarification should alleviate the need to include the control suggested.

Comment 4

Interpretation/clarification– None of these points are critical for the central claims of the paper, but I think needs some additional thought and clarification. Can the authors confirm the data in Fig7 is just the pre-infusion product from the Wilson Cancer Discovery paper? If it includes the post-infusion timepoints, there needs to be a lot more clarification and discussion because Bach2 is clearly not the same in antigen +/- conditions.

Response

Indeed, these were all pre-infusion products – we excluded any samples collected from patient peripheral blood. We have reinforced this in the text on **pages 8 and 15**.

Comment 4.1

I know the authors know that PD1 expression is not a definitive marker of ‘exhaustion’, but it conveniently lines up with some of the observations they are making. PD1 in most of these in vitro conditions is a readout of TCR signal strength and there are probably some much more interesting interpretations of the PD1 data in the paper unrelated to exhaustion. Something like Fig 1C where the strongest tonic signaling receptor (28-S) has the highest PD1 expression might be telling you something about how the receptor is acting. Or that in Fig 3J/K/L there is still 50% of the cancer left in the 28-S conditions and that has the most PD1 so its ot really a fair comparison. None of this is very important, but I would just suggest the authors think about their PD1 data in a little more detail than ‘exhaustion’.

Response

We could not agree more – PD1 expression is too often over-simplified, misinterpreted and misrepresented in our field. Candidly, we included these data since they have been requested by reviewers in the past. Our interpretation of PD1 expression relies on its association with the defective function of 22/28-S at the conclusion of manufacture (Figure 1g), which we do feel may be more suggestive of exhaustion (which we agree should be a functional and not phenotypic definition); regardless, we concur completely that the interpretation of these data is more complex. Our previous work has specifically pointed to the complexity in using PD1 as an “exhaustion” marker (Selli et. al., *Blood* 2023). If the Reviewer thinks appropriate, we are very open to adding language that highlights this complexity – we have not simply for sake of brevity and to avoid a discussion that may not be central to our conclusions.

Comment 4.2

I’m not sure if this is known, but will the e-coli derived protein tag be a target of rejection of the T cells if this is attempted in humans? rTTa is enough in mice so I would expect it might be an issue. This doesn’t take away from the good work, but probably needs some discussion.

Response

This is an issue we have been actively discussing with the company who has licensed the DHFR degron (Obsidian Therapeutics). They have begun initial clinical trials with a degron-linked cytokine, but one derived from humans. This issue is something we are acutely concerned about as we consider clinical translation ourselves. While there are some pre-clinical studies that can be done to evaluate immunogenicity, historically these have been poor predictors of humoral immunity in patients (murine-derived CARs targeting mesothelin are a clear example of this failure). We are considering several other avenues to enable BACH2 degradation (PROTACs, human carbonic anhydrase-derived degrons). We have text touching on this in the **Discussion**.

Reviewer #2

Comment 1 (major)

Since the study aims to establish BACH2 as a key transcriptional regulator induced by tonic signaling, a deeper epigenetic analysis is warranted. The identification of BACH2 as a transcriptional regulator in 22/BB-S tonic signaling cells is inferred from RNA sequencing data. To corroborate these findings, ATAC-seq should be performed to compare tonic versus non-tonic signaling CAR-T cells across the 2 different costimulatory domains (22/BB-S, 22/BB-L, 22/28-S, 22/28-L). Transcription factor (TF) motif enrichment analysis would help determine whether BACH2 emerges as a significantly enriched TF motif in 22/BB-S CAR-T cells. These analyses should also be conducted in both unedited and BACH2 knockout (KO) CAR-T cells to validate the results. Further, CHIP-seq is needed to identify direct BACH2 target genes and elucidate its regulatory network.

Response

The Reviewer is absolutely right; we inferred transcriptional regulation from RNAseq data and not classical motif analysis. To do this, we collected 22/BB-S, 22/BB-L, 22/28-S, 22/28-L and 22/BB-S+BACH2^{KO} cells at the end of manufacturing and performed ATACseq. chromVAR (Schep, *Nature Methods* 2017) analysis of these samples

revealed that the BACH2 motif was profoundly enriched in 22/BB-S T cells, with a deviation z-score of nearly 40. Re-assuringly, 22/BB-L and 22/BB-S+BACH2^{KO} cells had reduced and nearly equivalent BACH2 motif accessibility, while 22/28-S and 22/28-L had low BACH2 accessibility. These data are presented in **Figure 2c** and **Extended Data Fig. 1c**. Collectively with our transcriptional network inference data, *BACH2* transcript counts, BACH2 protein expression and BACH2 intranuclear localization studies, we believe we have comprehensively confirmed that 22/BB-S induces BACH2 activity.

Several previous studies have interrogated the BACH2 regulatory network using ChIP-seq (Tsukumo, *PNAS* 2013; Roychoudhuri, *Nature Immunology*, 2016; Hipp, *Nature Communications*, 2017). These are the reference GSEA datasets that allowed us to identify BACH2 as a regulator of the tonic 22/BB-S transcriptional program and were the datasets used to define the BACH2 regulon. At the Reviewer's suggestion, we undertook these studies ourselves in human CAR T cells. After several unsuccessful attempts to ChIP BACH2 we partnered with Mike Meers, faculty in the Department of Genetics at Wash U and a former post-doctoral fellow with Steve Henikoff – a pioneer in the development of novel chromatin interrogation techniques. Given the low abundance of BACH2 in human T cells and the absence of a validated BACH2 antibody for chromatin immunoprecipitation, we elected to perform cleavage under targets & release using nuclease (CUT&RUN). In addition to a higher sensitivity, this method is also far more precise than ChIP, enabling single base-resolution of binding sites. Unfortunately, after several months of troubleshooting with the Meers Lab, our most successful attempt to CUT&RUN BACH2 demonstrated a modest increase in BACH2 pull down as compared to IgG controls. Evaluation of the QC data through the R package ChIPQC (done by Dr. Meers) suggests that this is primarily a result of a weak antibody (there is only one commercially available α human BACH2 antibody) and likely compounded by the low abundance of BACH2 in human T cells. We were able to successfully perform CUT&RUN on the same cells using an antibody for cJun (**Figure 4h**), confirming that this technical barrier is likely related to the BACH2/antibody combination. We have included our list of conserved BACH2 binding targets (**Reviewer Table 1**), however do not feel it would be appropriate to include these in the manuscript as we would need extensive additional studies to confirm the validity of this list.

Only one study, to our knowledge, has successfully profiled BACH2 binding in human cells (Hipp et al., above). This study profiled BACH2 in B cells and appears to have been done in collaboration with a company (Active Motif Epigenetic Services). In light our technical constraints, despite engagement of expertise in chromatin profiling, and the several previous studies that have already elucidated the BACH2 regulatory network in mouse T cells, we hope the Reviewer will agree that continued troubleshooting of these studies would require significant time and effort without adding substantial support to the primary claims in this manuscript.

Comment 2 (major)

As BACH2 has already been established as a key transcriptional and epigenetic regulator of stem-like CD8+ T cells (PMID: 33574619), the novelty of this study lies in its connection to tonic signaling in CAR-T cells and the differential activity in CD28 and 41BB costim domains. A more detailed analysis of this association would strengthen the manuscript. Specifically, quantifying the level of tonic signaling at various time points during CAR-T cell manufacturing and correlating it with BACH2 expression at both transcriptional and proteomic levels would provide deeper insights.

Response

We appreciate the recognition of this novelty. Quantifying the level of tonic CAR signaling during manufacturing presents several technical and conceptual barriers. To ensure clarity, the tonic signal emanating from a CAR is, by definition, stable – it occurs through an intrinsic biochemical quality of the CAR that does not change and is not subject to external stimuli (as opposed to ligand-dependent signaling).

Human T cells require stimulation to remain alive in culture and this stimulation (either through the endogenous TCR using beads or through the CAR using antigen) overwhelms any tonic signal from the CAR. Measurement of

signaling at several time points of manufacturing would undoubtedly reveal signaling that results from the sledgehammer of bead stimulation. To better isolate the tonic CAR signal (and reduce the “noise” of CD3/CD28 bead stimulation), we chose the timepoint at which CAR T cells can remain alive but are temporally separated from stimulus – the end of manufacturing. Beyond this, quantification of tonic signal can only be done in comparison to another construct (there is no “scale” of tonic signaling to make an independent quantitative measurement). Identifying an association between tonic signal and BACH2 over a time course is further complicated by the fact that TCR stimulation is known to suppress BACH2 (Roychoudhuri, *Nature Immunology* 2016). This is also shown in our own data in which BACH2 levels rise in 22/BB-S cells the further these cells are in time from bead stimulation (**Figure 2f**). As such, we are faced with measurement of a signal that is diluted by the process of manufacturing and measurement of a protein that is degraded by activation. For these reasons, we believe that a clean analysis of tonic signaling and BACH2 over time of manufacturing is technically not feasible, and this is the rationale for evaluating these parameters at the end of manufacturing.

Lastly, and our apologies if we are misunderstanding the Reviewer’s comment, a direct comparison between quantity of signal from a CD28 or 41BB-based CAR is also very difficult, as these domains activate distinct signaling pathways. In theory, the biochemical property driving tonic signaling (scFv linker length) is the same for each CAR, but the emanating signaling is very likely both quantitatively and qualitatively different. We do not see this as a limitation, but simply a biological fact underlying our observation that tonic 41BB (and not CD28) induces BACH2. Again, our apologies if we misinterpreted this or any previous parts of the Reviewer’s comment.

Comment 3 (major)

In Vivo Studies need strengthening. The study relies on a single mouse model, and the observed improvement in tumor burden is modest. Additional in vivo models should be incorporated to substantiate the findings, including solid tumor models if possible.

Response

We completely concur and have extended our *in vitro* observations using the HA-GD2 CAR against neuroblastoma to an *in vivo* study. These data are now included in **Figure 6** and **Extended Data Fig. 8**. We established neuroblastoma in NSG mice and then delivered either untransduced, HA-GD2 or HA-GD2+BACH2^{DD} OFF, HA-GD2+BACH2^{DD} ON or HA-GD2+BACH2^{OE} (the same products that were evaluated in our initial *in vitro* studies). To more closely model clinically-relevant disease, we established neuroblastoma systemically, as opposed to more standard practice of subcutaneous tumor implantation and intratumoral CAR T cell delivery. We observed a very similar trend in tumor control to our *in vitro* studies, in which all three BACH2 overexpression products significantly enhanced anti-tumor efficacy, with the greatest control exhibited by BACH2^{DD} ON and BACH2^{OE}.

Intriguingly, and to our surprise, we found that mice treated with BACH2-engineered products began to lose weight ~3 weeks into treatment. A known complication of NSG models of T cell immunotherapy is xenogeneic graft-versus-host disease, characterized by hunching, fur loss, diarrhea and weight loss. Notably, our mice did not exhibit any symptom besides weight loss. As per our IACUC protocol, we sacrificed mice when they reached >15% weight loss. Necropsy revealed no signs of internal organ damage whatsoever however did reveal evidence of anorexia – the GI tracts of these mice were completely empty. The only other notable difference between NTD, HA-GD2 and control (no tumor) mice and HA-GD2+BACH2^{DD} OFF, HA-GD2+BACH2^{DD} ON or HA-GD2+BACH2^{OE} treated mice was a reduction in spleen size. We isolated cells from these spleens and found, as expected, significantly fewer total cells in these spleens but a significantly higher number of CAR+ cells. Memory phenotyping revealed trends consistent with our *in vitro* data in which BACH2^{DD} ON and BACH2^{OE} cells remained enriched for T_{CM} phenotypes.

While the ultimate cause of weight loss remains unclear, we speculate that one of three etiologies was responsible: (1) robust T cell engraftment leading to GVHD that only manifested as weight loss, (2) a hyperinflammatory cascade that led to systemic illness or (3) off-tumor CAR-mediated toxicity. Given that we did

not see this occur in our CD22 models (in which we deliver much higher doses of BACH2^{DD} and BACH2^{OE} CAR T cells) and have never seen this using this cell dose in this time frame in our previous extensive experience in NSG models of CD19 CARs, we speculate that the third etiology – off-tumor toxicity – is responsible. Previous reports have suggested that affinity-enhanced GD2 CARs can lead to neurotoxicity (Richman, *Cancer Immunology Research* 2018), which in some cases manifested as systemic wasting. We have discussed this observation with colleagues that have more extensive experience with the HA-GD2 CAR (Robbie Majzner, Boston Children’s Hospital and Evan Weber, Children’s Hospital of Philadelphia). While neither group has seen this specific form of toxicity, they confirmed that HA-GD2 CARs can induce a range of toxicities, speculated to be from promiscuous binding or high sensitivity for GD2 on healthy cells. We have several planned studies to further understand the mechanism of this toxicity that appears to be unique to GD2 CARs with enhanced functionality. Regardless, however, our data demonstrate that BACH2 manipulation not only enhances the anti-tumor function of the exhaustion-prone HA-GD2 CAR but may enhance its efficacy so much that it causes toxicity.

Comment 3 (major), continued

Moreover, key questions remain unanswered: What is the expansion and persistence of CAR-T cells in vivo? What is their phenotype over time? How does BACH2 expression change in vivo, and does it correlate with persistence and central memory formation?

Response

We completely understand how important these studies are to interpreting the *in vivo* data. We have since conducted experiments to analyze CAR T cells after infusion to mice. As outlined in Response to Comment 2.4 from Reviewer 1, serial tracking of CAR T cells in NSG peripheral blood is complex and often does not, in our hands, yield sufficient cells for detailed analysis (unless using the CD19 CAR). As such, we designed experiments to sacrifice animals at day 7 and day 13 and analyze splenic T cells. The data from these experiments are now included as **Figures 5m-p** and **Extended Data Fig. 7b**. Briefly, we found that 22/BB-S cells had the greatest expansion and persistence, with a trend (not statistically significant) of enhanced persistence of BACH2^{DD} OFF cells. BACH2 levels did not change over time *in vivo*: BACH2^{OE} retained high expression while BACH2^{DD} ON cells dropped expression to levels seen in BACH2^{DD} OFF cells. These data were expected since TMP is not given to mice and thus BACH2 is only stabilized during manufacturing (again, as in our *in vitro* studies; see **Figure 5j** for a schematic representation of BACH2 quantity in these cell products). We did find, again, a step-wise decline in CAR T cell differentiation that correlated with BACH2; BACH2^{DD} OFF cells were most successfully able to become T_{EM} cells, while BACH2^{OE} retained T_{CM} phenotypes. We also found that transgenic induction of BACH2, even at low levels induced in the BACH2^{DD} OFF cells, promoted CD4 lineages at the end of manufacturing, but that this skewing normalized over time in mice (**Extended Data Fig. 7f**) and *in vitro* (**Extended Data Fig. 5e**).

Specific Figure Comment 1:

Figure 1: The authors assess PD-1 expression, but it would be valuable to examine additional exhaustion markers such as LAG3, TIM3, and CTLA-4.

Response

Our apologies, we collected data on LAG3 and TIM3 throughout these studies but included only PD1 for brevity. We have now added these data to **Figure 1g**, which demonstrate very similar trends to the PD1 data. We have not routinely included CTLA4, primarily due to conflict with other colors in our panel, but could add transcript counts from our RNAseq data if the Reviewer believes it to be valuable.

Specific Figure Comment 2:

Figure 3g: Why does 22/BB-S not show high expression of memory-related genes (CCR7, LEF1, TCF7)?

Response

The reviewer brings up an important point that we clearly did not stress clarify in the manuscript text. While 22/BB-S does indeed have higher expression of memory-associated genes than 22/28-S, this is overwhelmed by the expression of these genes in 22/28-S+BACH2^{OE}. For example, 22/BB-S expresses 2.11x more *TCF7* than 22/28-S, but 22/28-S+BACH2^{OE} expresses 3.64x more *TCF7* than 22/BB-S (and thus 7.7x more than 22/28-S). These heatmaps reflect relative differences in gene expression as measured by z-score, which is scaled relative to the entire dataset and is not an absolute value; thus, if a specific sample (ie. 22/28-S+BACH2^{OE}) has a very high value expression of specific genes (ie. memory genes) the relative expression by other samples (22/BB-S) will look lower, even if it is still high.

Specific Figure Comment 3:

Figure 3n-q: Inclusion of 22/28-S as a control in these assays would help determine whether BACH2 KO specifically abrogates the advantage seen in 22/BB-S cells.

Response

We regret that we didn't include this control in the original experiments; we have now repeated these studies with the 22/28-S construct. These new data, presented in **Figures 3n-q**, clarify the impact of BACH2 loss in 22/BB-S cells. While 22/BB-S BACH2KO cells are less functional than 22/BB-S, they remain more effective than dysfunction-prone 22/28-S cells.

Specific Figure Comment 4:

Figure 3: Functional validation of BACH2 overexpression and KO CAR-T cells in rechallenge cytotoxicity assays would provide further insights into their long-term efficacy.

Response

Our apologies if this was not clear in our original submission; all *in vitro* experiments in **Figures 4-6** are rechallenge studies of BACH2^{OE} CAR T cells.

We did not originally conduct chronic challenge studies using BACH2^{KO} cells, since the functional impact of BACH2^{KO} was apparent even in short-term studies (**Figures 3n-q**) and did not require "stress" to uncover the functional impairment of BACH2 loss. At the Reviewer's request, we have performed chronic re-challenge studies using BACH2^{KO} 22/BB-S CAR T cells and found that loss of BACH2 has a modest impact on long-term efficacy of 22/BB-S cells. These data further support our hypothesis that endogenous BACH2 activity during CAR T cell manufacturing enhances fitness of 22/BB-S cells. We have included these data as **Reviewer Figure 1** and have not added them to manuscript, primarily because we have not done nearly the same level of in-depth interrogation of BACH2^{KO} as we did of BACH2^{OE} (which are the focus of the majority of studies in this manuscript). The reason for this is that BACH2^{KO} does not present a pathway to enhance function and these experiments using BACH2^{KO} cells simply support our hypothesis about BACH2's role. However, if the Reviewer believes these studies would be important to the narrative we are of course glad to include them in the manuscript.

Reviewer Figure 1. T cells were engineered to express either 22/BB-S or 22/28-S and some 22/BB-S cells were edited using a *BACH2* gRNA. Editing efficiency as measured by flow cytometry was >75%. Cells were then subjected to repeated stimulation by re-feeding of Nalm6, as described in the **Methods**, and function was measured over time.

Specific Figure Comment 5:

Figure 5b-j: Including 22/BB-S as a control would help to compare BACH2 expression levels between BACH2DD OFF and 22/BB-S, as well as their corresponding phenotypic and functional characteristics.

Response

We initially excluded 22/BB-S from the manuscript for simplicity of presentation; all experiments were conducted with both 22/BB-S and 22/28-S. We have now added these data back all analyses in **Figures 5b-i**.

Reviewer #3

Comment 1

The authors need to be clear when talking about tonic signaling from 4-1BB versus CD28 versus tonic signaling in general. This difference is unclear in multiple places in the paper (e.g. paragraph 2 of intro). Different receptors may have different types of tonic signaling and the authors need to be clear in their descriptions. A description of their previous findings on the short and long linker version of the anti-CD22-4-1BB CAR would be useful as well as signaling observed with the CD28 chimeras.

Response

We completely agree that these distinctions are of critical importance; our apologies that our language was not more precise. We have heavily modified the **Introduction** to now make this more specific and carefully outlined the differences between 41BB and CD28 tonic signaling.

Comment 2

In Fig 5m, what happens to the function if there is a titration of TMP?

Response

We did not originally perform titration of TMP, primarily based on our observation that even the “OFF” (which could more accurately be called “LOW” given the leakiness of the system) was sufficient to induce a robust T_{CM} differentiation during manufacturing and prevent the onset of exhaustion during chronic stimulation. We have since performed an *in vitro* dose titration study (testing TMP doses from 0.1 μ M, our previous “ON” condition, to 0.001 μ M). We confirm that, as seen in our immunoblot study in **Extended Data Fig. 5a**, BACH2 quantity is highly sensitive to TMP dose. Consistent with our “OFF” data, all BACH2 doses were sufficient to induce T_{CM} differentiation. Intriguingly, titrated dosing resulted in titrated activity, wherein CAR T cell expansion during chronic stimulation co-culture was directly linked to BACH2 quantity. Beyond this, we observed a titrated degree of transit from T_{CM} to T_{EM} states, mirroring our comparison between BACH2^{DD} OFF and ON, but now with greater degrees of granularity. We greatly thank the reviewer for suggesting these insightful studies, which add further evidence that CAR T cell function is highly sensitive to BACH2 dose. These data have been included in **Extended Data Figs. 7a-d**.

Comment 3

In the xenograft model, based on the data in Ext Data 4B, one would expect BACH2 levels to drop after a day after transfer, yet, it seems as if a subsequent reduction of BACH2 levels does not permit effector cell differentiation. Is it possible to check the BACH2 levels in the CAR-T cells several days after transfer in the xenograft model?

Response

The Reviewer correctly points out that we made an assumption about BACH2 quantity in CAR T cells after *in vivo* transfer based on our *in vitro* studies showing that, after manufacturing and withdrawal of TMP, BACH2 levels decline in BACH2^{DD} ON cells to the level seen in BACH2^{DD} OFF cells (**Figure 5b**). We have now performed studies to analyze CAR T cells on days 7 and 13 after transfer to mice. Indeed, we see a similar trend to our *in vitro* data wherein BACH2^{DD} ON cells have a drastic reduction in BACH2 but do not differentiate as effectively as BACH2^{DD} OFF cells (**Figures 5m-o**). These data, which are consistent with our *in vitro* chronic stimulation studies, further support our speculation that induction of high levels of BACH2 during manufacturing results in an irreversible restraining of effector differentiation even when BACH2 levels decline.

Comment 4

Based on the GD-2 model, the amount of Bach2 protein required to promote stemness and influence Tem appears to depend on the CAR-T construct (although the functionality is not tested). This makes the utility of the technique more questionable, although this is an important observation. Did the authors test the function of the GD-2 cells expressing BACH2?

Response

We regret that this was not clear in our original submission. **Figures 6c-g** all reflect functional evaluation of HA-GD2+BACH2^{DD} cells. We have now also added *in vivo* studies of HA-GD2 CAR T cells with transgenic BACH2, confirming that while all BACH2 conditions improve the function of HA-GD2 CAR T cells, high levels are associated with better control. These new studies are **Figures 6h-l** and include an exploration of the unexpected toxicity of anorexia and weight loss in HA-GD2+BACH2 treated mice. Please see the **Discussion** and our response to Comment 3 from Reviewer 2 above for more details about these toxicity studies.

Comment 5

What happens if BACH2-OFF is expressed in the 22/BB-S (and 22/BB-L) expressing cells? Does the small amount of BACH2 improve their subsequent function?

Response

The Reviewer raises a fascinating point that we had not previously considered. We performed a comparative study of WT, BACH2^{DD} OFF and BACH2^{OE} 22/BB-S and 22/BB-L cells and found that, like for 22/28-S, BACH2^{OE} significantly impaired long-term function. Like the Reviewer, we hypothesized, though, that BACH2^{DD} OFF in 22/BB-L cells would have induced similar function to 22/BB-S. Intriguingly, however, we found that low-level constitutive expression of BACH2 resulted in slightly worse function for both constructs. While we have not performed detailed analysis, we speculate that this observation reveals further nuance about: (1) the role of BACH2 in combating exhaustion versus inducing fitness, (2) the impact of BACH2 quantity during manufacturing in non-exhaustion-inducing, 41BB-based CARs, and (3) the impact of prolonged BACH2 activity. The relationship between BACH2 and 41BB is particularly intriguing and, based on these data and other data generated as part of other projects in our lab, we hypothesize that BACH2 may have fundamentally different activity in the absence of CD28 signaling. These observations have opened the door to a cadre of additional studies in our lab related to understanding the role of BACH2 outside of its ability to combat CD28-induced exhaustion. We have included these preliminary findings here as **Reviewer Figure 2** as we undertake studies to continue exploring this phenomenon. We hope the Reviewer understands their omission from this manuscript, as we hope to develop this with additional studies and report these findings in a distinct manuscript.

Reviewer Figure 2. 22/BB-S and 22/BB-L CAR T cells were engineered to co-express either BACH2^{OE} or BACH2^{DD} and then subjected to chronic re-stimulation with Nalm6.

Comment 6

Even in the context of the 22/28-S construct, did the authors try a titration of BACH2?

Response

We believe this comment is re-phrasing the comment raised in Comment 2 but for *in vitro* studies, which we have now performed. Our TMP titration studies are presented in **Extended Data Figs. 7a-d**.

Comment 7

In Fig 7d, BACH2 scores in CD4 Tcm correlates best with improved survival and no relapse —do the authors have further insight into this from other information on these cells. Are the CD4 cells cytolytic or do the effects appear secondary to/correlate with cytokine production that might promote CD8 cell function?

Response

We too were intrigued by these cells. While we are cautious to make conclusions based on sequencing (and not functional) data alone, further interrogation of these cells reveals that they did not express high levels of cytotoxicity-associated genes (granzymes, *PRF1*) or cytokines; in fact their expression of granzymes was lower than T_{CM} cells from patients who did relapse. The only marker we found to be significantly higher in these cells was *GATA3*. These findings are now included as **Figure 7e**. We are not entirely sure how to interpret these data but have added a brief discussion to the **Discussion** in light of recent data from Carl June and Jos Melenhorst identifying long-lived CD4 cells a decade after remission in a patient with CLL (Melenhorst, *Nature* 2022).

Comment 8

I think Ext Data 1c is miscited as 1e.

Response

Thank you for pointing this out – it was indeed miscited and we have corrected this error.